# Repurposing bunamidine hydrochloride as a potent antimicrobial agent targeting vancomycin-resistant *Enterococcus* membranes

Pengfei She,[1] Guanqing Huang,[2] Shaowei Guo,[2] Dan Xiao,[2] Yiqing Liu,[2] Mengna Li,[2] Lihua Lu,[2] Yelan Hong,[2] Yimin Li,[2] Linying Zhou,[2] Yong Wu[2]

**ABSTRACT** Vancomycin-resistant enterococci (VRE) are a major cause of hospital-acquired infections, with limited treatment options due to rising antibiotic resistance. Targeting bacterial membranes offers a promising alternative to conventional therapies. In this study, we investigated the antimicrobial activity and mechanism of action of bunamidine hydrochloride (BUN) against VRE. BUN exhibited potent bactericidal effects against both VRE and vancomycin-susceptible enterococci (VSE), with minimum inhibitory concentrations (MICs) ranging from 2 to 4 µg/mL. BUN significantly inhibited biofilm formation and eradicated biofilm-embedded persister cells. Mechanistic studies demonstrated that BUN disrupts bacterial membrane integrity, increasing permeability and depolarization, as confirmed by SYTOX Green staining, DiSC3(5) fluorescence analysis, and electron microscopy. Molecular dynamics simulations further revealed that BUN selectively interacts with phosphatidylglycerol, a key bacterial membrane phospholipid, leading to membrane destabilization. *In vivo* studies using murine infection models showed that BUN effectively reduced bacterial burden and promoted wound healing without notable toxicity. These findings highlight BUN as a promising antimicrobial agent with membrane-targeting activity against VRE. Its potent bactericidal action, low propensity for resistance development, and favorable *in vivo* efficacy suggest potential for therapeutic application in treating multidrug-resistant infections.

**IMPORTANCE** The global rise of vancomycin-resistant enterococci (VRE) poses serious challenges in clinical treatment due to limited therapeutic options and rapid resistance development. This study identifies bunamidine hydrochloride (BUN), a previously approved antiparasitic agent, as a potent membrane-targeting antimicrobial with rapid bactericidal activity against both planktonic and biofilm-associated VRE. By selectively interacting with bacterial phosphatidylglycerol, BUN disrupts membrane integrity and inhibits persister cells, while maintaining low cytotoxicity and high *in vivo* efficacy in murine infection models. Our findings highlight the potential of drug repurposing strategies to accelerate the discovery of effective antibiotics and provide a promising candidate for future clinical management of multidrug-resistant bacterial infections.

**KEYWORDS** bunamidine hydrochloride, vancomycin-resistant enterococci, bacterial membrane disruption, biofilm inhibition, phosphatidylglycerol targeting, drug repurposing

Antimicrobial resistance represents one of the most critical threats to global health in the 21st century, limiting therapeutic options and increasing mortality rates (1). Among the most concerning pathogens are vancomycin-resistant enterococci (VRE), which account for a substantial proportion of hospital-acquired infections (2, 3). VRE

**Peer Reviewers** Feifei Chen, Shanghai Institute of Materia Medica, Shanghai, China; Wenfang Lin, Institute of Urban Environment, Chinese Academy of Sciences, Xiamen, China

Address correspondence to Yong Wu, wuyong_zn@csu.edu.cn.

Pengfei She and Guanqing Huang contributed equally to this article. The author order was determined according to seniority.

The authors declare no conflict of interest.

See the funding table on p. 21.

is particularly problematic in immunocompromised patients and those undergoing invasive procedures, as its ability to acquire resistance genes and adapt to antibiotic pressure exacerbates treatment difficulties (4). The rise of VRE poses a serious clinical and public health threat. VRE infections are associated with prolonged hospital stays, increased medical costs, and higher mortality, especially among immunocompromised individuals (5). These infections are particularly problematic in hematological patients, where VRE colonization significantly increases the risk of subsequent infection and short-term mortality (6). Moreover, the horizontal transfer of vancomycin resistance genes—especially *vanA* and *vanB*—to other pathogens such as *Staphylococcus aureus* raises concerns about the potential emergence and spread of vancomycin-resistant *S. aureus* (7), further highlighting the urgency of developing alternative therapies.

Conventional antibiotics target essential bacterial biological processes such as cell wall synthesis, RNA translation, or DNA replication (8). However, resistance mechanisms, including enzymatic degradation, efflux pumps, and target modifications, have rendered many of these therapies increasingly ineffective (9, 10). Consequently, research efforts are now exploring alternative targets, particularly the bacterial cell membrane (11, 12). This vital structure is essential for maintaining cellular integrity, energy production, and metabolic regulation (13), making it a highly attractive target. Strategies aimed at disrupting membrane permeability offer promising solutions to overcome resistance and combat antimicrobial resistance (14–16).

Bacterial membranes comprise various components critical for survival, including phospholipids such as phosphatidylglycerol (PG), phosphatidylcholine (PC), phosphatidylethanolamine (PE), and cardiolipin (CL), as well as integral membrane proteins (17). Phospholipids like PG are abundant in bacterial membranes yet scarce in mammalian cells, providing selective targets for antimicrobial development (14, 18). Studies widely reported the high antimicrobial efficacy of antimicrobial peptides, polymyxin derivatives, and dendritic small molecules by disrupting membrane integrity and inducing bacterial cell death (19). These findings demonstrated the potential of membrane-targeting agents as effective alternatives to conventional antibiotics (20, 21).

Drug repurposing has gained momentum as a cost-effective strategy to address antimicrobial resistance (22). By identifying novel antimicrobials in drugs approved or under investigation, repurposing efforts largely reduce the development timelines and bypass early safety evaluations (23). Recent studies have highlighted the untapped potential of this approach in combating resistant pathogens, demonstrating its utility in expanding therapeutic options (24).

Drug repurposing has emerged as an effective strategy to accelerate the development of novel antimicrobials by leveraging the established pharmacological and safety profiles of approved compounds. For instance, auranofin, originally developed for the treatment of rheumatoid arthritis, has been shown to exhibit potent antibacterial activity through inhibition of thiol-redox enzymes (25). Among the various drug classes investigated, antiparasitic agents have attracted particular attention due to their structural diversity and intrinsic bioactivity. For example, niclosamide displays broad-spectrum antibacterial effects by disrupting bacterial energy metabolism and redox homeostasis (26); tafenoquine exerts strong bactericidal activity through membrane targeting (27); Oxyclozanide and closantel, two salicylanilide-based anthelmintics, display strong activity against Gram-positive pathogens, likely through membrane depolarization (28). Nitazoxanide, another clinically used antiparasitic drug, inhibits anaerobic metabolism and alters membrane function (29). Collectively, these agents highlight the antimicrobial potential of repurposed antiparasitics, especially those acting on bacterial membranes or energy systems. In this context, bunamidine hydrochloride (BUN), a classical antiparasitic compound, was selected for further investigation to evaluate its antibacterial efficacy and underlying mechanism against VRE.

BUN was initially developed as an antiparasitic agent (30). However, its effects on bacterial pathogens remain underexplored. This study aims to evaluate the *in vitro* and *in vivo* antimicrobial potential of BUN against VRE and further explore its underlying

molecular mechanisms, thereby contributing to the development of novel therapeutic strategies against VRE in clinical settings.

## MATERIALS AND METHODS

### Bacterial strains and cell culture

The bacterial strains used in this study, including VRE and VSE isolates, are summarized in Table S1. All *Enterococcus* strains were cultured in brain-heart infusion (BHI) broth at 37℃. Human skin fibroblast (HSF) and HaCaT keratinocyte cells were maintained in high-glucose DMEM supplemented with 10% fetal bovine serum (FBS) and 1% penicillin-streptomycin under standard conditions (37℃, 5% $CO_2$). BUN and other chemicals were purchased from MedChemExpress (New Jersey, USA) and dissolved in dimethyl sulfoxide (DMSO) or deionized water for stock solutions.

### Antimicrobial susceptibility test

The MIC of BUN and other antimicrobials was determined using the standard microdilution method according to Clinical and Laboratory Standards Institute guidelines (2024). Briefly, bacteria in the logarithmic phase were diluted to a final concentration of ~1 $\times 10^6$ colony-forming units (CFU)/mL in Mueller-Hinton (MH) II broth (Solarbio, Beijing, China). Fifty microliters of the bacterial suspension was added to a 96-well plate (Corning Costar, USA) with an equal volume of serially diluted antimicrobials. After incubating at 37℃ for 16–20 h, the bacterial growth turbidity was assessed by measuring the optical density at 630 nm ($OD_{630}$) with a microplate spectrophotometer (Bio-Rad iMark, USA). The MIC was defined as the lowest concentration that inhibited visible bacterial growth. For the minimum bactericidal concentration (MBC) determination, 10 µL of the bacterial suspension in each well was transferred onto sheep blood agar plates, and the MBC was defined as the lowest concentration that achieved 99.9% viable bacterial cells killing on the agar after overnight incubation (31).

### Bactericidal dynamics

Log-phased *E. faecium* and *E. faecalis* cultures were diluted in MH broth containing BUN or other antimicrobial agents to a final concentration of ~1 $\times 10^6$ CFU/mL. DMSO (0.1%, vol/vol) was used as a control. All the groups were incubated at 37℃ at 180 rpm, and an aliquot of the bacterial suspension was collected at the time points of 0, 2, 4, 8, 12, and 24 h, respectively, to measure the bacterial viability. Quantitative CFU enumeration was performed through 10-fold serial dilution in sterile saline, followed by plating on sheep blood agar plates (32).

### SYTO9/PI probe-based live/dead staining

Log-phased *E. faecalis* was centrifuged, washed twice with 1× PBS (pH ≈ 7.2), and resuspended to $OD_{630}$ = 0.1 in the presence of BUN at indicated concentrations. After 2 h incubation at 37℃, the cells were washed again and stained with 10 µM of SYTO9/propidium iodide (PI) at the ratio of 1:1 (vol/vol). After incubation in the dark for 15 min, the residual probes were removed by PBS washing, and the bacterial viability was observed using a confocal laser scanning microscope (CLSM; LSM800, ZEISS, Germany) (33), where the scale bars were generated automatically by the acquisition software. Unless otherwise specified, all CLSM imaging experiments in this study were performed using *E. faecalis* ATCC 51299.

### Resistance induction assay

The potential for resistance development by BUN was assessed by both stepwise and single-step resistance induction assays, respectively. For stepwise resistance induction, the MIC of BUN was measured as described above on the first day. The next day,

the bacterial suspension from the well of 1/2× MIC was diluted 1:1,000 with fresh MH broth and further exposed to serially diluted BUN for MIC determination after 24-h incubation. The values of MIC were consecutively recorded for 15 days. For single-step resistance induction, overnight *E. faecalis* cultures were adjusted to ~$1 \times 10^8$ CFU/mL in sterile saline, and 100 µL of the suspension was spread onto MH agar plates in the presence of various concentrations of BUN. After 48-h incubation at 37℃, the colonies with drug-resistant mutations were counted, and the mutation frequency was calculated as the ratio of resistant CFUs to the initial inoculum. In addition, rifampin (RFP) was used as a positive control for both assays (34).

## Post-antibiotic effect assay

To evaluate the post-antibiotic effect (PAE), *E. faecium* and *E. faecalis* were exposed to BUN at concentrations of 1–4× MIC in Mueller-Hinton (MH) broth (5 mL per tube, initial inoculum ~$5 \times 10^6$ CFU/mL). MH broth containing 1% DMSO served as a negative control. After incubation at 37 ℃ with shaking (180 rpm) for 1 h, bacterial suspensions were washed by 1:1,000 dilution in fresh MH broth to eliminate residual antimicrobial activity. Subsequently, samples were collected at 0, 2, 4, 8, 12, and 24 h, serially diluted, and plated for CFU enumeration. Vancomycin (VAN) and daptomycin (DAP) at 10× MIC were included as positive controls for comparative analysis. For comparative purposes, the MICs of VAN and DAP against *E. faecalis* ATCC 51299 and *E. faecium* U101 were determined as 32 µg/mL and 4 µg/mL, respectively. Unless otherwise specified, VAN and DAP were used at 10× MIC in all related comparative assays, including those shown in Fig. 2K and 3D.

## Ultrastructural observation by electron microscopy

Log-phased *E. faecalis* was washed and re-suspended in the 1× PBS in the presence of 5× MIC of BUN or 0.1% DMSO (control). After incubation at 37℃ for 1 h, the bacterial cells were fixed with 2.5% glutaraldehyde overnight, and the ultrastructural changes were observed by using scanning electron microscopy (SEM) and transmission electron microscopy (TEM) (HITACHI, Tokyo, Japan), respectively (35).

## Membrane potential determination by DiSC3(5)

Mid-log phased *E. faecalis* cells were suspended in HEPES buffer (5 mM, pH 7.2, supplemented with 5 mM glucose and 100 mM KCl) to an $OD_{630} = 0.05$. And the bacterial suspension was incubated with 2 µM of DiSC3(5) for 1 h in the dark. Ten microliters of BUN at specified concentrations was added to 90 µL of the bacterial suspension, with melittin (MLT; 8 µg/mL, corresponding to its MIC against *E. faecalis* ATCC 51299) as a positive control and 0.1% DMSO as a negative control. The fluorescence intensity was measured at the excitation/emission wavelengths of 622/670 nm, respectively, using a multimode plate reader (PerkinElmer EnVision, UK) (36).

## Membrane permeability monitoring by SYTOX Green

Mid-log phased *E. faecalis* cells were washed and resuspended in 1× PBS to $OD_{630} = 0.05$ in the presence of 2 µM of SYTOX Green. After incubation in the dark for 30 min, 50 µL of the bacterial suspension with 50 µL of BUN (final concentration of 1/4–2× MIC) was added to a black 96-well plate. Melittin (8 µg/mL) and 0.1% DMSO were used as positive and negative controls, respectively. The fluorescence intensity was detected at the excitation/emission wavelengths of 485/525 nm, respectively (37).

## Cell viability determination by calcein-AM/PI staining

HSF cells were cultured in DMEM with 10% FBS and 1% penicillin-streptomycin at 37℃ and 5% $CO_2$, then treated with 1× MIC of BUN for 24 h, harvested, and washed with PBS. Next, the cells were incubated with 2 µM of Calcein-AM and 8 µM of PI for 30 min in

the dark. Cellular morphology was imaged using an inverted fluorescence microscope (Zhongxian, Beijing, China), and the fluorescence intensity was analyzed with ImageJ (Version 1.53t, National Institutes of Health, USA) (38).

## Cytotoxicity determination by CCK-8 assay

HSF and human keratinocyte (HaCaT) cells were maintained in Dulbecco's Modified Eagle Medium (DMEM; Gibco, Thermo Fisher Scientific, USA) supplemented with 10% FBS (Gibco, Thermo Fisher Scientific, USA) at 37°C in a humidified atmosphere containing 5% $CO_2$. Cells in the logarithmic growth phase were seeded into 96-well plates at a density of $5 \times 10^3$ cells/well in 100 µL of complete medium and allowed to adhere for 24 h. Subsequently, the medium was replaced with fresh medium containing the indicated concentrations of BUN. Cells treated with 0.1% DMSO served as the vehicle control. After a 24- h exposure, 10 µL of Cell Counting Kit-8 (CCK-8; Dojindo, Japan) solution was added to each well and incubated for an additional 3 h at 37°C. Absorbance was measured at 450 nm ($A_{450}$) using a microplate reader, and cell viability was calculated relative to the control group (39).

## Apoptosis detection

HSF cells in the logarithmic phase were treated with 4 µg/mL of BUN for 24 h, and 1% DMSO was used as a control. Then, the cells were stained with annexin V-FITC/PI as described by the manufacturer's specifications. Apoptosis was determined by using flow cytometry (BD, USA) (40).

## Biofilm inhibition assay

Overnight bacterial cultures were 1:100 diluted in BHI broth supplemented with 2% (wt/vol) glucose (BHIg) in a 96-well plate in the presence of serially twofold diluted antimicrobial agents. After static incubation at 37°C for 24 h, the supernatant in each well was removed and gently washed with 1× PBS. For crystal violet staining, 0.15% (wt/vol) crystal violet solution was added to each well, after incubation at room temperature for 15 min, excess crystal violet was removed by PBS washing. One hundred microliters of ethanol was added to each well for 20 min incubation to dissolve the biofilm-bound crystal violet. The absorbance at 560 nm ($A_{560}$) was measured by using a microplate reader. For the XTT assay, XTT (0.2 mg/mL) and phenazine methosulfate (PMS; 0.02 mg/mL) were mixed in 1× PBS, and 100 µL of the solution was added to the PBS-washed biofilms after the treatment. The absorbance at 490 nm ($A_{490}$) was measured after incubation in the dark for 3 h (41).

## Biofilm eradication assay

Overnight cultures of enterococci were 1:100 diluted with BHIg and dispensed into a 96-well plate. After static incubation at 37°C for 24 h, the supernatant was removed, and the wells with adhered biofilms were gently washed with PBS. Fresh BHIg broth containing serial twofold dilutions of antimicrobial agents was added. The plates were incubated statically for an additional 24 h, followed by PBS washing again. The biofilm biomass and viability were determined by crystal violet staining and the XTT method as described above, respectively (41).

## Biofilm visualization by CLSM

The biofilms were cultured in six-well plates with a sterile coverslip in each well. After the treatment as described above, planktonic cells were removed, and the biofilms were washed with PBS. Then, the biofilms were stained with 10 µM of SYTO9/PI for 15 min in the dark and visualized by using the CLSM (LSM800, Zeiss, Jena, Germany). The fluorescence intensity was quantified by ImageJ (42), where the scale bars were generated automatically by the acquisition software.

## Persister killing assay

Mid-logarithmic phase cultures of *E. faecalis* ATCC 51299 grown in BHI broth were adjusted to ~1 × $10^8$ CFU/mL. Aliquots (100 µL) were transferred into 96-well polystyrene plates, sealed, and incubated at 37 °C under humid conditions for 24 h to allow biofilm formation. Planktonic cells were removed by washing twice with 1× PBS, and the biofilms were subsequently exposed to 100 µL BHI broth containing RFP (100× MIC) to induce the formation of biofilm-embedded persister cells. After incubation at 37°C for 24 h, the biofilms were washed twice with PBS, and the remaining adherent cells were resuspended in 100 µL PBS followed by sonication for 5 min to disrupt the biofilm matrix. The resulting cell suspension was transferred into 200 µL PBS containing BUN at various concentrations or comparator antibiotics (vancomycin, daptomycin). The mixtures were incubated statically at 37°C, and aliquots were collected at 1- h intervals for CFU enumeration over a total of 4 h (43).

## Drug combination

The checkerboard dilution method was used to determine the combinational antimicrobial effects between BUN and conventional antibiotics. Briefly, *Enterococcus* cultures in mid-log phase were diluted with MH broth to ~1 × $10^6$ CFU/mL. Serially twofold diluted BUN was added horizontally (50 µL/well) to a 96-well plate, while an equal volume of the conventional antibiotic was added vertically. Then, the plate was incubated at 37°C for 16–24 h, and the MIC values for single and combined use were recorded. The fractional inhibitory concentration index (FICI) was calculated as follows (44):

$$\text{FICI} = \frac{\text{MIC}A(\text{combination})}{\text{MIC}A(\text{alone})} + \frac{\text{MIC}B(\text{combination})}{\text{MIC}B(\text{alone})}$$

## Molecular dynamics

The phospholipid membrane models were constructed using CHARMM-GUI software, where the bacterial membrane was composed of 168 DOPC (1,2-dioleoyl-sn-glycero-3-phosphocholine) and 72 DOPG (1,2-dioleoyl-sn-glycero-3-phosphoglycerol) molecules (DOPC:DOPG = 7:3), while the mammalian membrane consisted of 168 POPC (1-palmitoyl-2-oleoyl-sn-glycero-3-phosphocholine) and 72 cholesterol molecules (POPC:cholesterol = 7:3). These models were solvated and charge-neutralized in preparation for molecular dynamics (MD) simulations. The MD simulations were performed using Gromacs 2019.6. Initially, energy minimization was conducted with the steepest descent method to eliminate atomic clashes. This was followed by 100 ps of NVT (constant number of particles, volume, and temperature) and NPT (constant number of particles, pressure, and temperature) equilibration simulations at a controlled temperature of 300 K (room temperature). The simulation for both bacterial and mammalian phospholipid membranes ran for 500 ns, with configurations saved every 100 ps. Visualization of the simulation was carried out using Gromacs embedded tools and VMD. Analysis of the trajectories indicated that BUN primarily interacted with the 7DOPC/3DOPG bilayer through diffuse hydrophobic contacts between its aromatic/alkyl moieties and lipid tails, while hydrogen bonding events were rare and transient. Because hydrophobic interactions are spatially dispersed and non-directional, and the contribution from hydrogen bonding was minimal, these specific interactions were not explicitly annotated in the structural visualizations to maintain rigor and avoid overinterpretation (45).

## Growth inhibition assay

Mid-log phase *Enterococcus* was diluted with MH broth to the final concentration of ~1 × $10^6$ CFU/mL. Then the tubes were added with twofold diluted antimicrobials. DMSO (1%) served as control. After static incubation for 16–18 h, the bacterial cells were washed and centrifuged. Then, the pellets were re-suspended in 1 mL of XTT (0.2 mg/mL) with

PMS (0.2 mg/mL). After being incubated in the dark for 3 h, the bacterial suspension was transferred to a 96-well plate and $A_{490}$ was measured (46).

## Lysis assay

Mid-log phase *Enterococcus* cultures were adjusted to an $OD_{630} \approx 0.55$ in saline. BUN or lusutrombopag (LP) was added to the bacterial suspension at a final concentration of 32 µg/mL. LP served as the bacteriolytic control, and DMSO (1%, vol/vol) was used as the negative control. After incubation at 37°C and 180 rpm, the suspensions were collected at the time points of 0, 1, and 2 h, respectively, and the bacterial turbidity was measured by $OD_{630}$ (47).

## Mouse wound infection model

All animal-related procedures were approved by the Ethics Committee of the Affiliated Changsha Hospital of Xiangya School of Medicine, Hunan, China (NO. CSU-2024-0280). Female ICR mice (6–8 weeks old, 23–27 g) were obtained from Hunan SJA Laboratory Animal Co., Ltd. To establish an *Enterococcus faecalis* wound infection model, dorsal fur was removed using a chemical depilatory agent before inoculating a 50 µL suspension containing $1 \times 10^8$ CFU onto a 5 mm full-thickness skin defect. One hour post-infection, for each sampling time point (days 1, 3, and 7), mice were randomly assigned into three treatment groups (*n* = 9 mice per group per time point): 2% BUN cream, 2% fusidic acid (FD) cream as a positive control, or 2% (vol/vol) DMSO cream as a negative control. Treatments were applied topically twice daily for 7 days, with all wounds covered by a protective membrane. Wound healing progression was monitored daily via imaging and diameter measurements. At each designated time point, six mice per group were sacrificed for bacterial burden analysis following homogenization in 1 mL sterile saline, while the remaining three mice per group were fixed in 4% paraformaldehyde for histological examination. Tissue sections underwent dehydration, paraffin embedding, and subsequent staining with hematoxylin-eosin (H&E), Masson's trichrome, and Giemsa stains (Servicebio, Wuhan, China). Immunofluorescence staining was performed to assess IL-6, IL-1β, and TNF-α expression using a Nikon E100 fluorescence microscope (Japan) (48).

## Mouse subcutaneous abscess model

ICR female mice (6–8 weeks old, 23–27 g) were obtained from Hunan SJA Laboratory Animal Co., Ltd. Dorsal fur was removed using a chemical depilatory agent prior to the experiment. *Enterococcus faecalis* ATCC 51299 was cultured to the logarithmic phase, washed with sterile saline, and resuspended. A 100 µL suspension containing $1 \times 10^7$ CFU was subcutaneously injected into the dorsal region to establish the infection model. One hour post-infection, mice were randomly assigned to two groups (*n* = 6) and received subcutaneous administration of either 30 mg/kg BUN or 1% (vol/vol) DMSO (vehicle control). Twelve hours post-infection, the mice received a second injection of the same drug subcutaneously. After 24 h of treatment, the mice were euthanized, and infected skin tissue was collected for homogenization and bacterial count 0 (49).

## *In vivo* toxicity

ICR female mice (6–8 weeks old, 23–27 g) were assigned to two groups (*n* = 6) and received subcutaneous injections of either a vehicle solution (1% [vol/vol] DMSO) or BUN (30 mg/kg) twice, with a 12-h interval. After 24 h of treatment, mice were euthanized, and infected skin, along with major organs (heart, liver, spleen, lungs, and kidneys), was collected. Tissue specimens were fixed in 4% paraformaldehyde for 24 h, followed by dehydration through a graded ethanol series and xylene clearance before paraffin embedding and sectioning. Histopathological assessments were performed using H&E

staining for internal organs, while Masson's trichrome staining was applied specifically to wound skin tissues (Servicebio, Wuhan, China) (50).

## Pharmacokinetic analysis

Pharmacokinetic (PK) studies of BUN were performed in female ICR mice (6–8 weeks old, 23–27 g; Hunan SJA Laboratory Animal Co., Ltd., China). Mice were randomly assigned into three groups ($n = 6$ per time point) and received BUN via intravenous (i.v.; 10 mg/kg), subcutaneous (s.c.; 30 mg/kg), or intraperitoneal (i.p., 30 mg/kg) administration. Blood samples (~100 μL) were collected from the retro-orbital sinus at predetermined time points (0.083, 0.25, 0.5, 1, 2, 4, 8, and 12 h) into heparinized tubes and immediately centrifuged at 4°C (4,000 × $g$, 10 min) to obtain plasma. Plasma samples were stored at ~80°C until analysis. BUN concentrations in plasma were quantified using a validated LC–MS/MS method. PK parameters, including maximum plasma concentration ($C_{max}$), time to reach $C_{max}$ ($T_{max}$), elimination half-life ($t_{1/2}$), area under the concentration–time curve from time zero to infinity ($AUC_{0~\infty}$), and absolute bioavailability (F%), were calculated using non-compartmental analysis with WinNonlin software (version 8.3, Pharsight, USA).

## Statistical analysis

Data are presented as mean ± standard deviation (SD) from at least three independent experiments. Statistical analysis was performed using one-way ANOVA. ns: not significant; *$P < 0.05$; **$P < 0.01$; ***$P < 0.001$; ****$P < 0.0001$.

## RESULTS

### Potent bactericidal effects of BUN against VRE

The chemical structure of BUN comprises a benzimidazole moiety scaffold and an aromatic ring system containing two nitrogen atoms, which is known to enhance biological activity. Meanwhile, BUN features a long alkyl chain, a characteristic that enhances lipophilicity, facilitating bacterial membrane penetration (Fig. 1A). By microbroth dilution assay, BUN demonstrated strong antimicrobial activity against both VRE and vancomycin-susceptible enterococci (VSE), with MIC and MBC values ranging from 2 to 4 μg/mL and 2 to 8 μg/mL, respectively (Table 1). Notably, its bactericidal efficacy was also verified across varied clinical isolates (Table 1). Notably, BUN exhibited similar inhibitory activity against certain *Staphylococcus epidermidis* and *S. aureus* strains, including MSSA and MRSA with MIC of 4–8 μg/mL, supporting its broad antimicrobial potential against Gram-positive pathogens. However, given the urgent clinical challenge posed by VRE, characterized by high multidrug resistance, hospital transmission, and limited treatment options, we prioritized VRE as the primary focus of this study. Different from agents with predominantly bacteriostatic activity, such as LP (34), BUN exhibited significant lytic activity against VRE within a 2-h treatment (Fig. 1B). The XTT assay demonstrated a concentration-dependent growth inhibition of BUN against VRE (Fig. 1C). Next, the antimicrobial activity of BUN was compared in different culture media. As shown in Fig. 1D, the inhibitory effect was comparable between MH and BHI broth, indicating that the choice of culture medium had minimal influence on the observed activity. Furthermore, kinetic studies revealed the concentration-dependent bactericidal effects of BUN. Complete elimination of viable cells in both strains was achieved at 1× MIC within 2 h (Fig. 1E). As expected, live/dead bacterial cell determination using SYTO9/PI probes showed a significant increase in dead cells in the BUN-treated group after 2-h incubation (Fig. 1F). Consistently, quantitative fluorescence analysis also demonstrated the fast bactericidal activity of BUN against VRE (Fig. 1G). However, the checkerboard dilution assay indicated non-synergistic antibacterial activity between BUN and conventional antibiotics (Fig. S1).

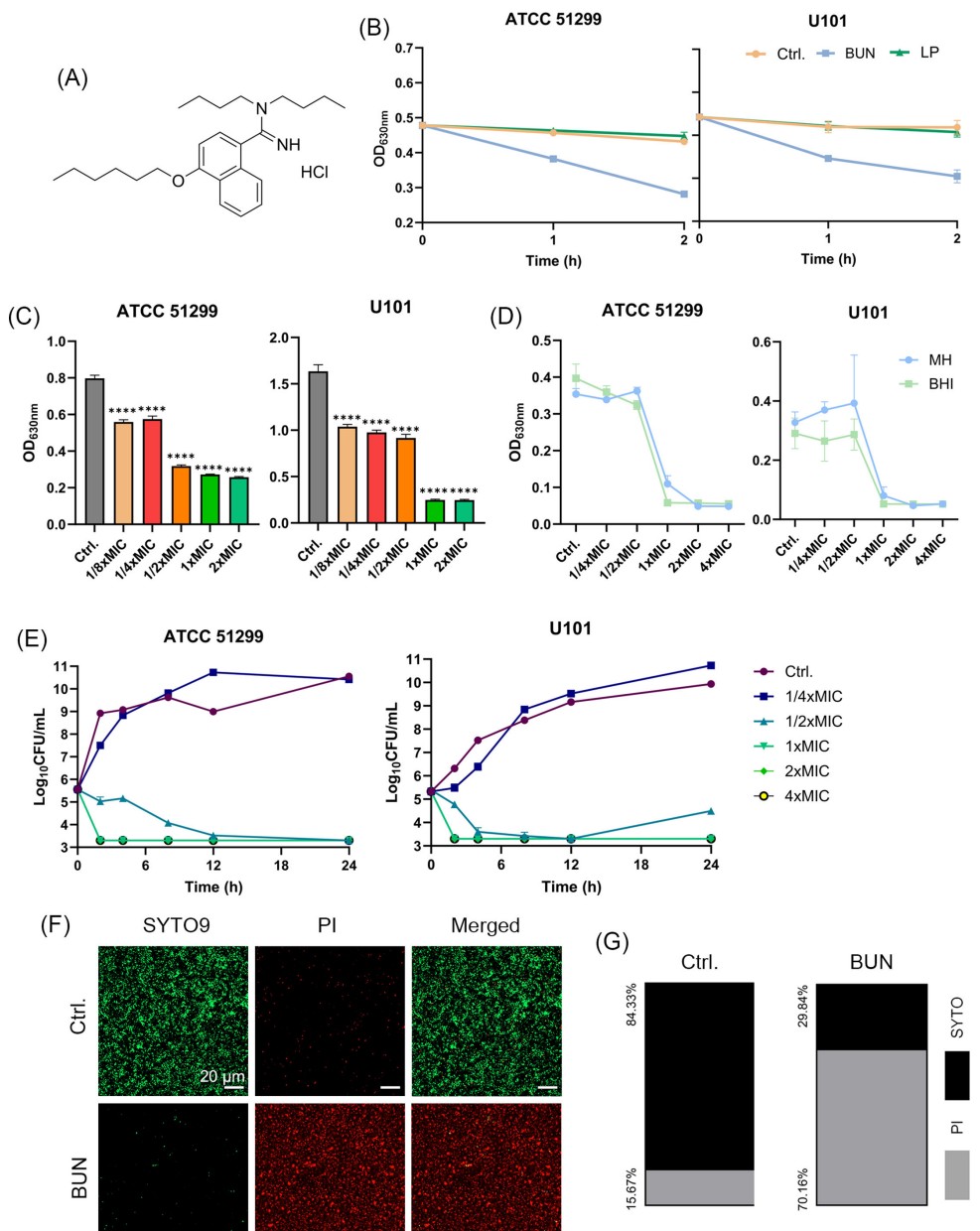

**FIG 1** Rapid bactericidal activity of BUN. (A) Chemical structure of BUN. (B) Lytic activity of BUN against VRE. DMSO (1%, vol/vol) and Lusutrombopag were served as negative and bacteriostatic controls, respectively. (C) Quantification of bacterial optical density at different BUN concentrations using XTT reduction assay. (D) Concentration-dependent growth inhibition activity of BUN against VRE in the presence of MH or BHI broth. (E) Time-killing curves of BUN against VRE. (F) Representative CLSM images of *E. faecalis* ATCC 51,299 cells stained with SYTO9/PI after treatment with 1× MIC of BUN for 2 h. Scale bars: 20 µm (G) Quantitative fluorescence analysis of SYTO9/PI-stained cells. Revised to include a color legend indicating SYTO and PI staining.

## Bactericidal activity of BUN against biofilms and persister cells

The antibiofilm activity of BUN against VRE was assessed using crystal violet staining and XTT reduction assay to determine biofilm biomass and metabolic activity, respectively. Crystal violet staining indicated that BUN significantly reduced biofilm biomass formation at the concentration of 4 µg/mL, while XTT assay confirmed a corresponding decrease in biofilm viability (Fig. 2A). For biofilm eradication, crystal violet staining

showed significant matrix degradation at 4 µg/mL and further biomass reduction at 8 µg/mL, while the XTT assay mirrored these findings, demonstrating a substantial decline in metabolic activity at 8 µg/mL (Fig. 2B). To account for potential differences in planktonic growth, biofilm biomass data were normalized as biofilm formation units (BFU = $A_{560}/A_{600}$). The normalized results showed a consistent trend, with significant differences observed from 4 µg/mL onwards (Fig. S2). These findings suggest that BUN effectively inhibits biofilm formation at 4 µg/mL and disrupts mature biofilms at 8 µg/mL, supporting its potential as a therapeutic agent against VRE biofilm-associated infections. To further characterize the antibiofilm effects of BUN, we extended our analysis to CLSM, allowing a more detailed visualization of structural changes within the biofilm matrix. Representative two-dimensional (2D) CLSM images of biofilms treated with BUN at 4 µg/mL (biofilm inhibition) and 8 µg/mL (biofilm eradication) are shown in Fig. 2C and E, respectively. In the biofilm inhibition assay, a notable reduction in biofilm biomass and an increased proportion of PI-stained impaired cells were observed at 4 µg/mL, indicating early-stage structural disruption (Fig. 2C). For pre-formed biofilms, treatment with BUN at 8 µg/mL resulted in extensive biofilm eradication, with a marked reduction in biofilm thickness and a significant increase in PI-stained dead cells (Fig. 2E).

To quantitatively assess these structural alterations, fluorescence analysis of SYTO9/PI staining was performed across multiple fields of view. The proportion of PI-positive cells was significantly higher in the BUN-treated biofilm inhibition group compared to the control (Fig. 2D), and a similar trend was observed in the biofilm eradication group at 8 µg/mL (Fig. 2F). Given the promising results from 2D CLSM imaging, three-dimensional (3D) CLSM imaging was employed to further elucidate the spatial organization of BUN-treated biofilms. Compared to the untreated control, biofilms treated with BUN at 4 µg/mL exhibited substantial inhibition in biofilm development (Fig. 2G), while treatment at 8 µg/mL led to extensive biofilm disruption (Fig. 2I). Corresponding fluorescence quantifications further confirmed these findings, revealing significantly higher proportions of PI-positive cells in treated biofilms (Fig. 2H and J).

The bactericidal effects of BUN against biofilm-embedded persister cells were further evaluated by CFU enumeration. As shown in Fig. 2K, BUN at 1× MIC effectively inhibited the growth of persister cells within the biofilms for both *E. faecalis* ATCC 51299 and *E. faecium* U101. Notably, BUN achieved complete killing of biofilm-embedded persister cells within 1 h at the concentration of 2× MIC. In contrast, VAN and DAP only exhibited limited antimicrobial activity against persister cells even at the concentration of 10× MIC. Together, these findings demonstrated that BUN not only effectively inhibited VRE biofilm formation but also exhibited strong bactericidal activity against pre-existing biofilms and biofilm-related persister cells.

## Resistance development assessment of BUN

The potential of BUN to induce bacterial resistance was assessed using stepwise and single-step assays, respectively. As shown in Fig. 3A, in the stepwise induction assay, BUN did not induce significant MIC values elevation, suggesting a low propensity for resistance selection, while RFP exhibited a substantial MIC values increase. To further evaluate the emergence of resistance mutations occurring under high concentration antimicrobials selection pressure, a single-step induction assay was conducted. As shown in Fig. 3B, although RFP treatment led to the obvious occurrence of resistant mutations, the BUN-treated group exhibited no resistant colonies detected. In consistence, the resistance mutation frequency quantification indicated that the BUN-treated group exhibited remarkably lower frequency of resistant mutation reduction than the RFP-treated group (Fig. 3C).

Beyond its favorable resistance-inducing profile, BUN also exhibited a prolonged PAE. As shown in Fig. 3D, the CFU counting recovery time was approximately 4 h after treatment with 1× MIC of BUN in both VRE strains. In contrast, the 10× MIC concentrations of DAP and VAN showed much less effect, with bacterial recovery similar to the control group, where a large bacterial load persisted. Taken together, these results

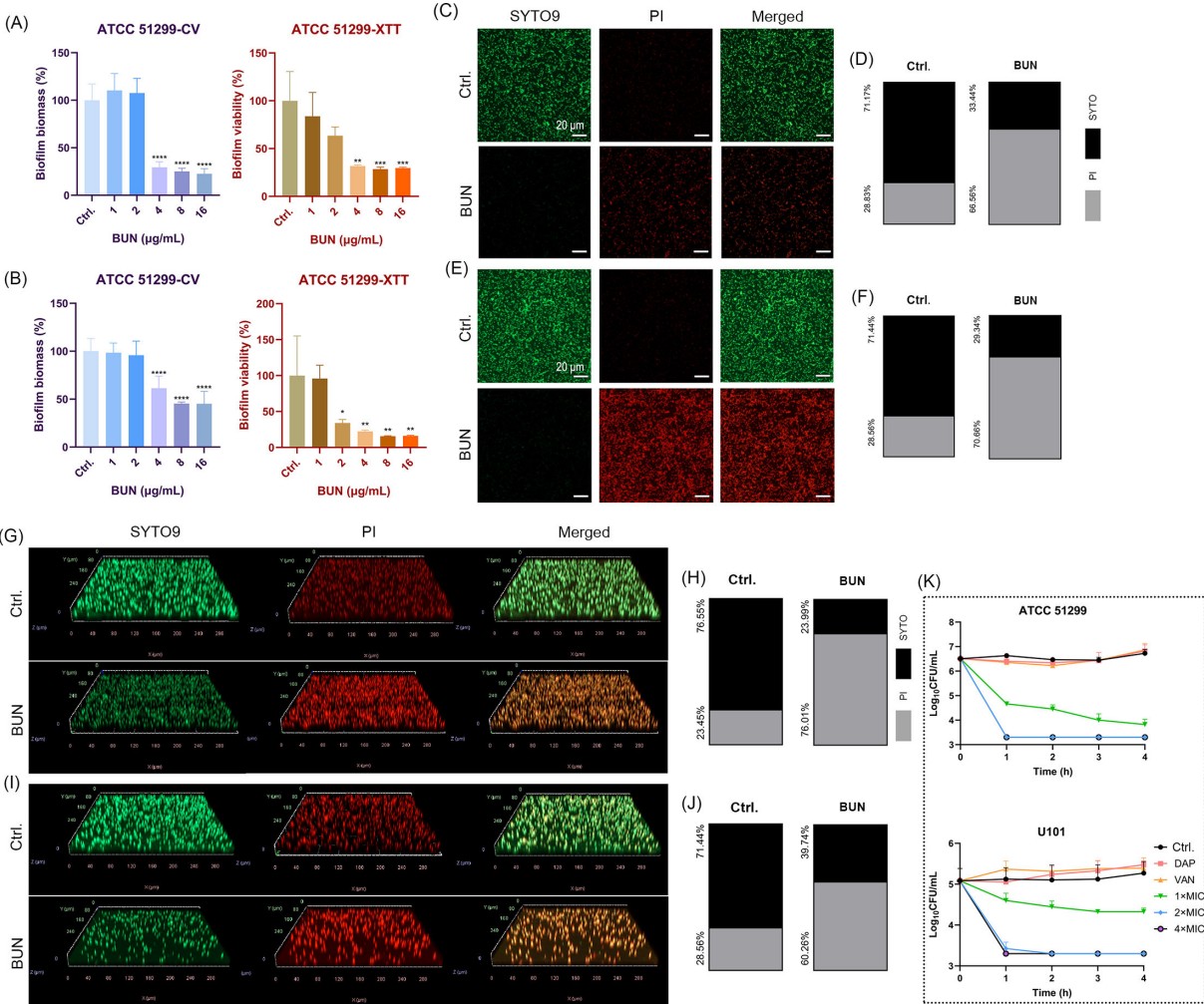

**FIG 2** Antimicrobial activity of BUN against VRE biofilms and its persister cells. All CLSM images in this figure were acquired from *E. faecalis* ATCC 51299 biofilms. (A and B) Inhibitory (A) and eradicating (B) effects of BUN against *E. faecalis* ATCC 51299 biofilms, evaluated by CV staining for biomass and XTT assay for metabolic activity, respectively. (C) Representative two-dimensional (2D) CLSM images of SYTO9/PI-stained biofilms in the biofilm inhibitory assay after 4 µg/mL of BUN treatment. Scale bars: 20 µm (D) Quantitative fluorescence analysis of the biofilm inhibition. Revised to include a color legend indicating SYTO and PI staining. (E) Representative CLSM images in the biofilm eradication assay after 8 µg/mL of BUN treatment. Scale bars: 20 µm (F) Quantitative fluorescence analysis of the biofilm eradication. (G) Three-dimensional (3D) CLSM images of SYTO9/PI-stained biofilms in the biofilm inhibitory assay. (H) Quantitative fluorescence analysis of the 3D biofilms in the biofilm inhibitory assay. Revised to include a color legend indicating SYTO and PI staining. (I) Representative CLSM images in the biofilm eradication assay. (J) Quantitative fluorescence analysis of the 3D biofilms in the biofilm eradication assay. (K) Bactericidal kinetics of BUN against biofilm-embedded VRE persister cells, with vancomycin (VAN;10× MIC) and daptomycin (DAP; 10× MIC) as conventional antibiotic controls. *$P < 0.05$, **$P < 0.01$, ***$P < 0.001$, ****$P < 0.0001$.

highlight the low resistance risk and prolonged PAE of BUN, suggesting that BUN may offer a long-term advantage in combating drug-resistant VRE-related infections compared to conventional antibiotics.

## Mechanism of action

SEM and TEM were employed to observe the morphological alterations in BUN-treated bacterial cells. SEM images displayed cell shrinkage, collapse, and loss of membrane integrity, while TEM revealed obvious membrane disruption, including rupture, detachment from the cell wall, and leakage of cytoplasmic contents. Some cells exhibited irregular intracellular structures and cytoplasmic disorganization (Fig. 4A). These morphological changes suggest that the antimicrobial mechanism of BUN may be

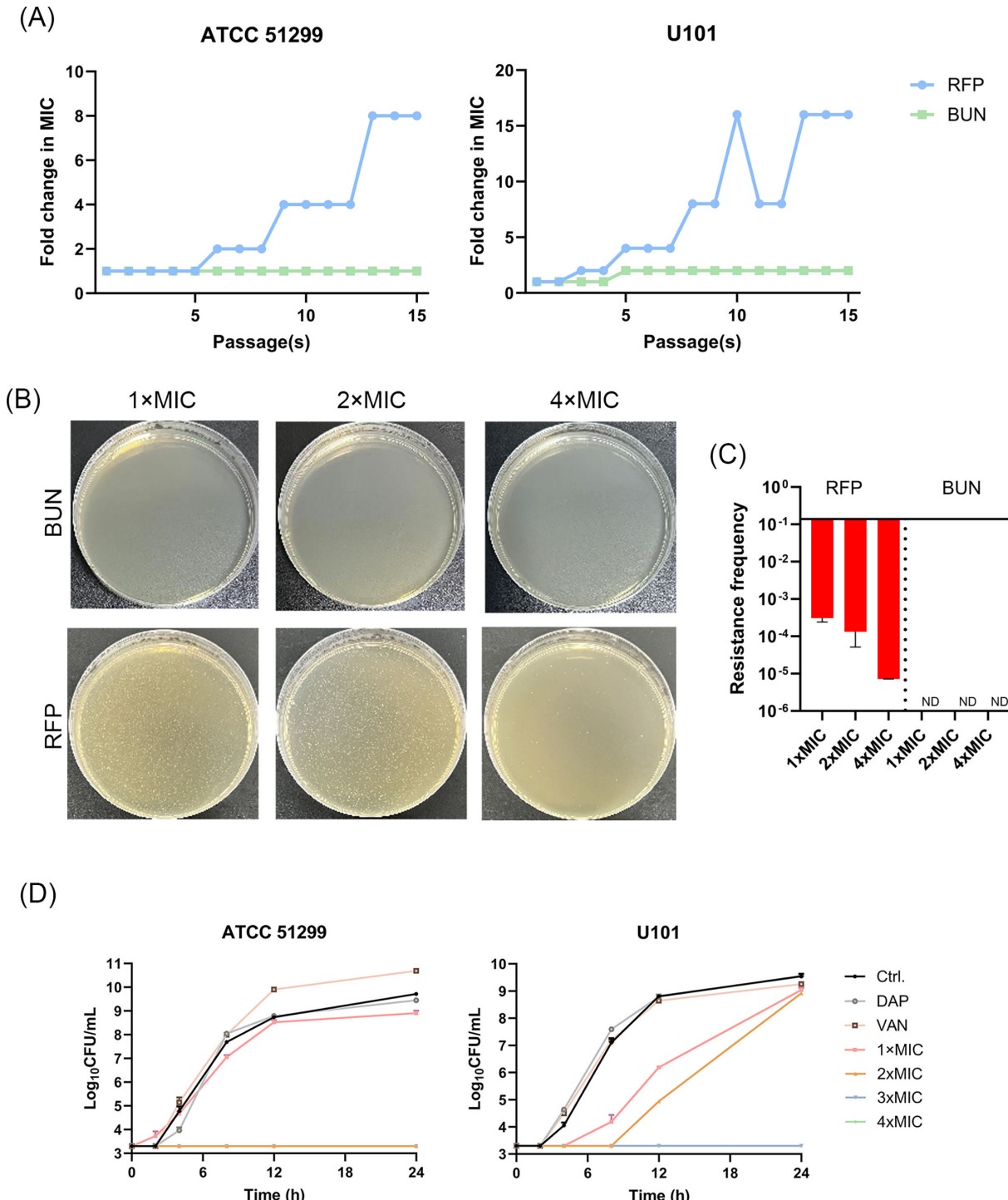

**FIG 3** Low resistance induction potential of BUN. (A) Stepwise resistance induction by BUN over 15 days, with RFP as a control. (B) Single-step resistance induction against *E. faecalis* ATCC 51299 by BUN at concentrations ranging from 1 to 4× MIC. RFP was used as a control. (C) Quantification of resistance frequency in the single-step assay. (D) PAE analysis of BUN against VRE of *E. faecalis* ATCC 51299 and *E. faecium* U101, with VAN (10× MIC) and DAP (10× MIC) as conventional antibiotic controls.

mediated by bacterial cell membrane disruption. Next, we investigated the impact of BUN on membrane permeability using the SYTOX Green probe. As shown in Fig. 4B, BUN treatment induced a concentration-dependent increase in SYTOX Green fluorescence at 0.25–2× MIC, indicating early membrane permeability changes. CLSM observation

further confirmed these findings, with fluorescence microscopy images showing enhanced SYTOX Green uptake in BUN-treated cells compared to the control (Fig. 4C). By using DiSC3(5), a membrane potential-sensitive dye, BUN treatment caused a significant fluorescence reduction in a concentration-dependent manner, suggesting membrane depolarization (Fig. 4D). To complement these findings, the PI probe was used to assess membrane integrity, revealing a substantial increase in fluorescence at 1–2× MIC, indicative of significant membrane damage (Fig. 4E). Notably, melittin (MLT), a known pore-forming peptide, showed strong PI fluorescence but relatively weak SYTOX Green and DiSC3(5) signals in comparison. This discrepancy may be attributed to the rapid and extensive pore formation induced by MLT, which allows immediate influx of larger dyes like PI while disrupting membrane potential to a lesser extent. In contrast, BUN induces progressive membrane depolarization and permeability alterations, reflecting a distinct, non-lytic mode of membrane disruption that primarily interferes with functional integrity rather than direct perforation. Together, the early membrane damage detected by SYTOX Green, the membrane potential changes observed with DiSC3(5), and the severe membrane impairment and cell death observed with PI provide a comprehensive view of the membrane-disrupting bactericidal mechanism by BUN.

To elucidate the specific molecular target of BUN in bacterial membranes, we performed competitive inhibition assays with various phospholipid components. Among them, BUN exhibited the obvious antagonistic antimicrobial activity with PG (Fig. 4F), while PE, PC, or CL only exhibited moderate or no interaction with BUN (Fig. S3). These results suggest that BUN primarily interacts with PG, thereby disrupting membrane integrity. Growth curves of *E. faecalis* ATCC 51299 further demonstrated the inhibitory activity of PG against the antimicrobial effects of BUN (Fig. 4G).

To further investigate the interaction of BUN with bacterial cell membranes at the molecular level, we conducted MD simulations using 7DOPC/3DOPG and 7POPC/3Cholesterol bilayers to model bacterial and mammalian membranes, respectively. DOPG was selected as a functional analog of PG, a major bacterial membrane phospholipid, while DOPC and POPC were used as representative phospholipids for PC-containing membranes in mammalian cells. Although DOPC and POPC differ slightly in their fatty acid composition, both share similar biophysical properties, making them suitable models for PC-rich environments. As shown in Fig. 4H, BUN rapidly adsorbed to the surface of the 7DOPC/3DOPG bacterial membrane model, forming hydrogen bonds with the phosphate groups of phospholipids, which facilitated stable binding. Over time, BUN gradually inserted into the membrane and stabilized in the outer leaflet of the lipid bilayer. In contrast, BUN remained on the surface of the 7POPC/3Cholesterol mammalian membrane model, with no evidence of penetration. These findings highlight the preferential interaction of BUN with bacterial-like membranes rich in PG analogs. Further analysis revealed that BUN induced noticeable changes in membrane thickness within the 7DOPC/3DOPG bilayer, whereas the 7POPC/3Cholesterol bilayer remained largely unchanged (Fig. 4I). Distance fluctuation measurements demonstrated that BUN rapidly approached the 7DOPC/3DOPG bilayer and stabilized at 2.01 ± 0.47 nm after 300 ns, whereas its distance to the 7POPC/3Cholesterol bilayer remained relatively unchanged at an average of 4.98 ± 0.53 nm (Fig. 4J). Hydrogen bonding analysis further confirmed that BUN formed multiple hydrogen bonds with DOPG-containing membranes, reinforcing its stable integration, whereas only moderate hydrogen bonding was observed in the cholesterol-rich mammalian membrane model (Fig. 4K). Consistently, binding energy analysis revealed that BUN exhibited a much stronger affinity for bacterial-like membranes, as reflected by a significant decrease in binding energy over time (Fig. 4L).

Molecular interaction mapping further demonstrated that the aromatic ring and hydrophobic chain of BUN strongly interact with the hydrophobic tails of phospholipids, stabilizing BUN within the membrane while disrupting lipid packing, leading to membrane destabilization (Fig. 4M). Additionally, BUN established robust hydrophobic interactions with surrounding phospholipids, further supporting its stable membrane binding and its disruptive effect on bacterial membranes (Fig. 4N). In all, the binding of

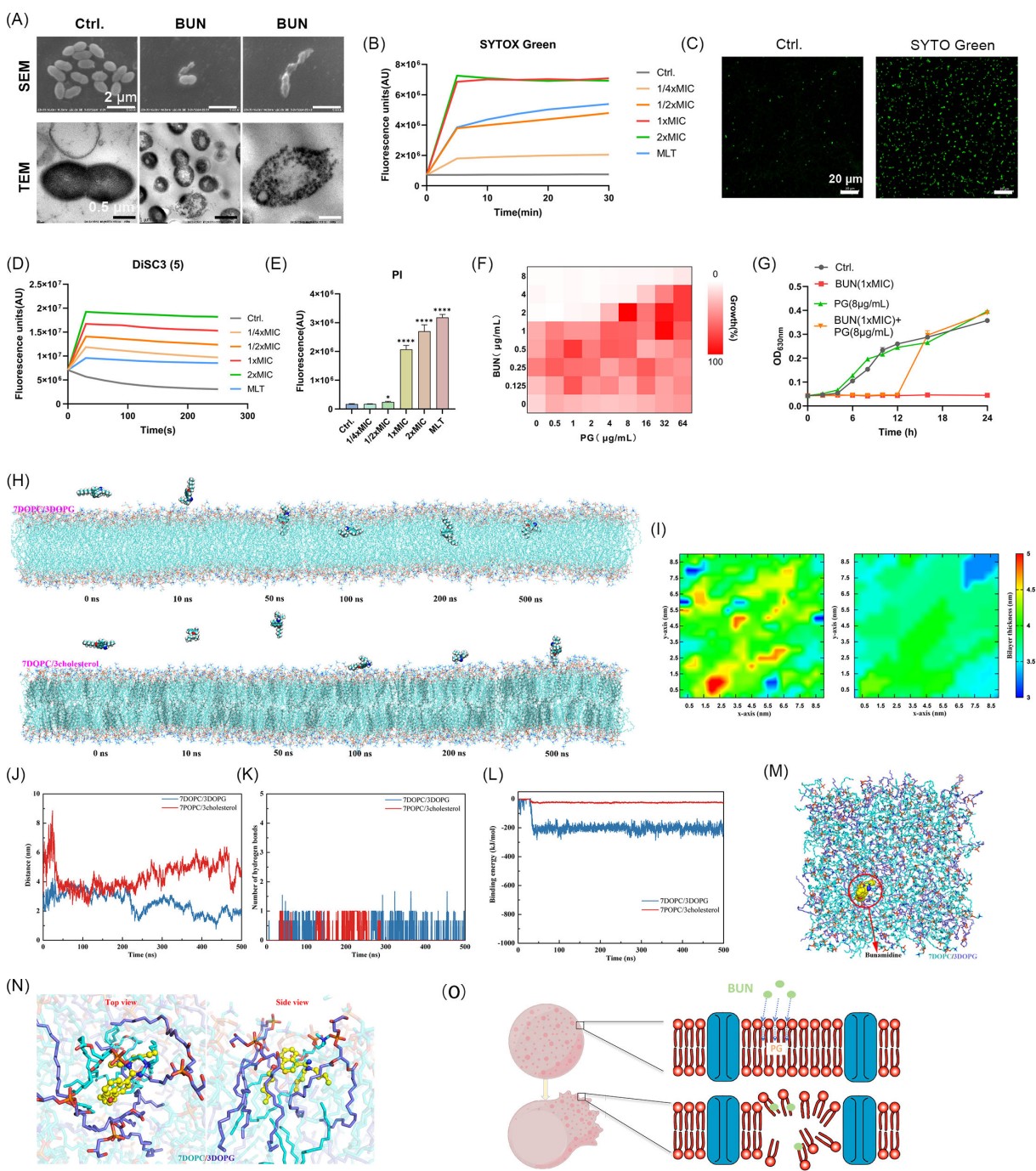

**FIG 4** Bacterial cell membrane-disrupting effects of BUN. (A) SEM and TEM images of *E. faecalis* following BUN treatment. Scalebars: 2 µm for SEM and 0.5 µm for TEM. (B) *E. faecalis* membrane permeability assessment with SYTOX Green probe. (C) Representative images of *E. faecalis* stained with SYTOX Green with or without BUN treatment. Both groups were adjusted to the same bacterial density before treatment. Scale bars: 20 µm. (D) Membrane potential depolarization after BUN treatment was measured using the DiSC3(5) probe. (E) Membrane integrity evaluation by PI uptake assay. (F) The interaction between BUN and PG was determined by the checkerboard assay. (G) Time-growth curves of *E. faecalis* in the presence of BUN (1× MIC) and PG alone or in combination. (H) Representative MD images of the interaction between BUN with bacterial or mammal membranes. (I) Membrane thickness simulation analysis of the lipid bilayers after being treated with BUN. (J) Distance fluctuations between BUN and the membranes over the simulation period. (K) Hydrogen bonding quantification between BUN and the lipid molecules. (L) Binding energy analysis of BUN with different membranes. (M) Top-down view of BUN localization within the lipid bilayer. (N) Molecular interaction map between BUN and lipid molecules on the top and side views, respectively. MD simulations showed predominant hydrophobic contacts and minimal, transient hydrogen bonding; specific interactions were not annotated to maintain rigor and avoid overinterpretation. (O) Mechanism diagram of BUN-induced bacterial membrane disruption. *$P < 0.05$,****$P < 0.0001$.

BUN to the membrane surface leads to destabilization of the lipid bilayer, resulting in increased permeability and ultimately membrane disruption (Fig. 4O).

## Biocompatibility and cytotoxicity evaluation

The IC50 values of BUN in HSF (>29.5 µg/mL) and HaCaT cells (>14.8 µg/mL) were significantly higher than its MIC values against VRE, indicating a favorable safety profile with low cytotoxicity at therapeutic concentrations (Fig. 5A). Flow cytometry was performed on HSF cells to assess the effect of BUN on apoptosis (Fig. 5B). As we expected, there was no significant difference in the apoptosis rate between the control group and BUN treatment group at 1× MIC after 24 h of incubation (Fig. 5C). The 24-h exposure duration reflects a standard acute toxicity assessment commonly used *in vitro*, providing a preliminary safety profile relevant to short-term therapeutic use. Additionally, Calcein AM/PI staining also showed no obvious apoptosis in the BUN-treated group (Fig. 5D). These findings supported the conclusion that BUN is unlikely to cause significant damage to human cells at therapeutic concentrations.

## *In vivo* efficacy against VRE infections

The PK parameters of BUN in mice demonstrated favorable *in vivo* performance (Fig. S4; Table S3). The plasma concentration-time profiles indicated that BUN exhibited a half-life ranging from 5.06 to 7.85 h across different administration routes. The $T_{max}$ of BUN was relatively short (0.11–0.51 h), suggesting rapid absorption. BUN exhibited relatively good bioavailabilities of 27.80% and 58.95% for s.c. and i.p., respectively.

To evaluate the antimicrobial efficacy of BUN *in vivo*, a mouse subcutaneous abscess model was first established. Infected mice were subcutaneously injected with 30 mg/kg BUN or 1% (vol/vol) DMSO (negative control), and bacterial burden was quantified post-treatment. As shown in Fig. 6A, BUN treatment significantly reduced bacterial loads in abscess tissues, with bacterial counts decreasing by approximately 1 $\log_{10}$ CFU/mL compared to the vehicle group. Similarly, CFU counting revealed a substantial decrease in viable bacterial colonies following BUN treatment (Fig. 6B). Given that VRE-associated wound infections pose great clinical challenges, we further performed a mouse wound infection model. Representative images in Fig. 6C illustrated the progression of wound healing over 7 days, with BUN treatment visibly promoting wound closure compared to the vehicle group. Consistently, the wound area exhibited a clearer downward trend over time in the BUN or FD-treated groups compared with the vehicle group (Fig. 6D). Meanwhile, CFU enumeration was performed. As shown in Fig. 6E, BUN treatment significantly reduced bacterial burden in wound tissues, with a decrease of approximately 2 $\log_{10}$ CFU/mL on day 1 compared to the vehicle group. In addition, inflammatory cytokine levels were measured to assess the host immune response. IL-1β expression was significantly lower in the BUN group, while no significant difference was observed; we still found the decreasing tendency of IL-6 and TNF-α levels (Fig. 6F). Histological analysis further supported these findings: H&E staining revealed reduced inflammatory infiltration in the BUN-treated wounds, while immunohistochemical staining for IL-1β, IL-6, and TNF-α highlighted a less intense immune response compared to the vehicle group (Fig. 6G). Masson's trichrome staining indicated improved collagen deposition, suggesting enhanced tissue repair. These results collectively demonstrate that BUN effectively reduces bacterial burden, mitigates inflammation, and promotes wound healing in VRE-infected wounds.

Next, the potential systemic toxicity of BUN was assessed. Blood routine parameters, including white blood cell (WBC) count and neutrophil percentage (Neu%), showed no significant differences between the BUN and vehicle groups. Similarly, serum biochemical markers, including alanine aminotransferase (ALT), also showed no significant differences. Histological analysis of major organs (myocardium, liver, spleen, lung, kidney) using H&E staining revealed no observable pathological damage in the BUN-treated group compared to the control. Additionally, Masson's trichrome staining of skin tissues showed normal collagen deposition patterns, further supporting the safety of

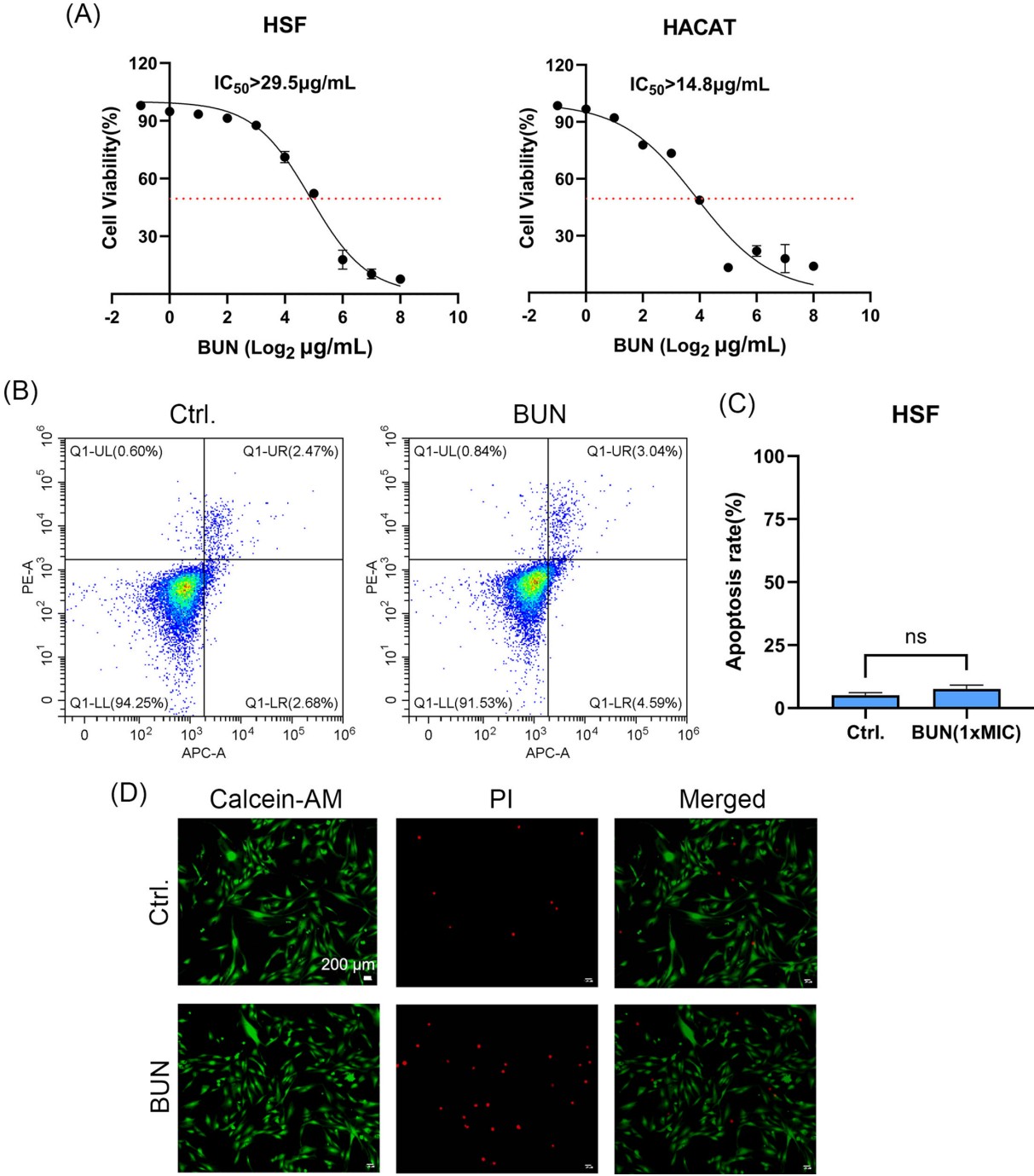

**FIG 5** Cytotoxicity and apoptosis analysis in human cells. (A) Cytotoxicity of BUN against HSF and HaCaT cells assessed by CCK-8 assay. (B) HSF cells apoptosis determination by Annexin V-FITC/PI staining after being treated with BUN for 24 h. (C) Quantification of the apoptotic rate. (D) HSF cells apoptosis determination by calcein-AM/PI staining. The cells were treated or untreated with 4 µg/mL BUN for 24 h. Scale bar: 200 µm. ns: no statistical significance.

BUN treatment. These findings highlight the minimal *in vivo* systemic toxicity of BUN (Fig. S5).

## DISCUSSION

Given the increasing prevalence of VRE, a major pathogen responsible for multidrug-resistant infections in clinical settings (8, 51), we identified BUN, an antiparasitic agent, exhibiting significant antibacterial activity against VRE with low propensity for resistance

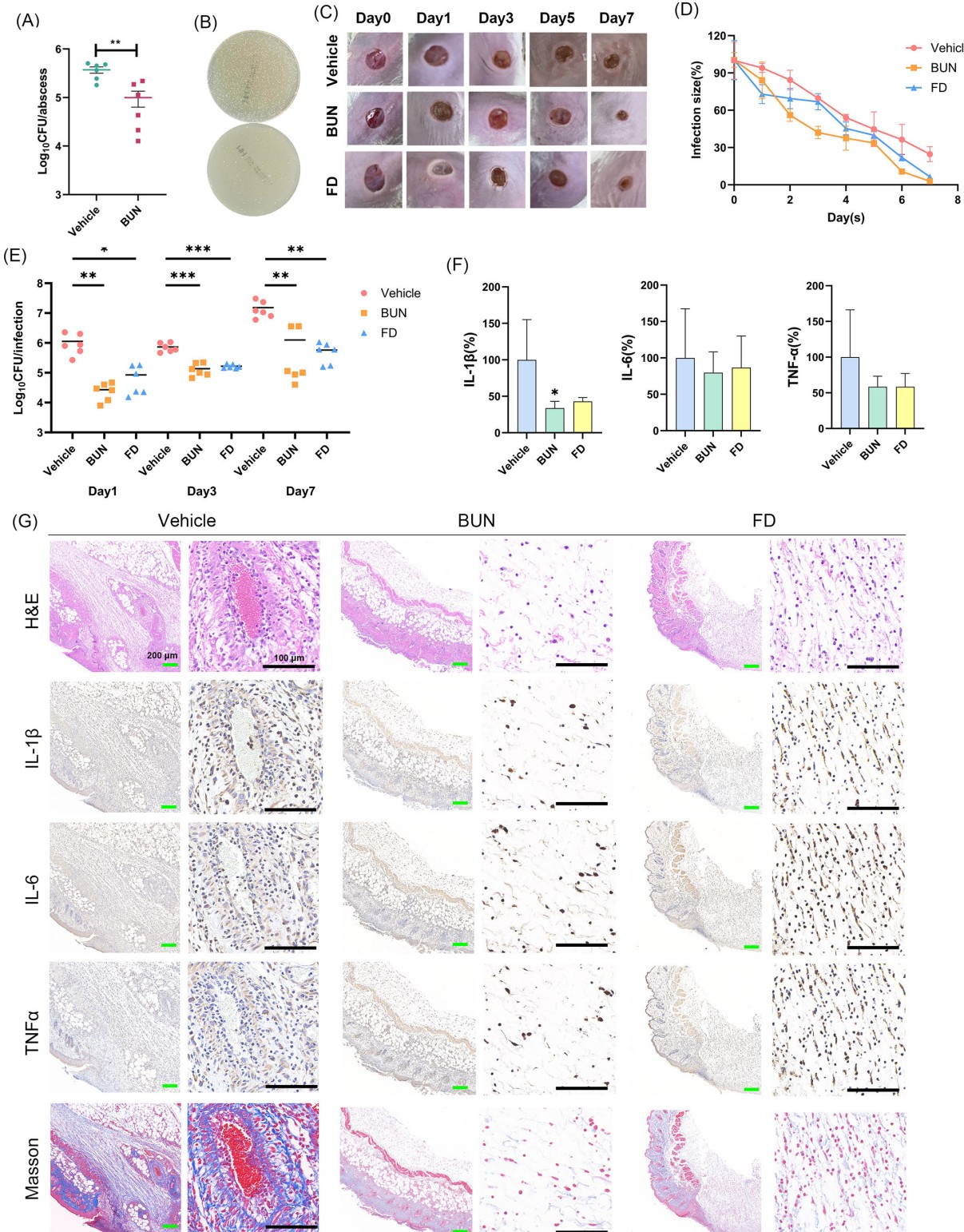

**FIG 6** BUN promotes infection clearance and wound healing *in vivo*. (A) Bacterial burden in the subcutaneous abscess model after being treated with 30 mg/kg BUN or 1% (vol/vol) DMSO for 24 h. (B) Representative bacterial CFU counting images of the abscess homogenates. (C) Representative images of the wound healing progression over 7 days in the mice treated with vehicle (2% DMSO), 2% BUN, or 2% FD. (D) Quantitative analysis of the wound area over time. (E) Bacterial viable loads in the wound tissues on days 1, 3, and 7, respectively. (F) Quantification of the inflammatory cytokines (IL-1β, IL-6, and TNF-α) in the wounds on day 7. (G) Histological and immunohistochemical analyses of the wound from different treatment groups. H&E staining was used to assess tissue

Fig 6 (Continued)

morphology and inflammatory infiltration, while the immunohistochemical staining was performed to evaluate the inflammatory responses of IL-1β, IL-6, and TNF-α. Giemsa staining was used to observe bacterial cells, and the Masson staining was used for collagen deposition observation. Slight edge artifacts may appear in a few histological panels due to unavoidable tissue unevenness during sectioning and mounting. These subtle features are not visible in the original unprocessed images but may be accentuated under certain image enhancement conditions. Such artifacts do not affect pathological interpretation. Green scale: 200 µm. Black scale bars: 100 µm. *$P < 0.05$, **$P < 0.01$, ***$P < 0.001$.

development. Mechanistic studies indicate that BUN exerts its bactericidal effects by targeting the bacterial cell membrane, leading to increased membrane permeability, depolarization, and subsequent cell lysis (14, 52). And this disruption is mediated mainly through interactions with PG, a key phospholipid component of the VRE membrane (14). In addition, BUN effectively exerted antimicrobial activities against VRE biofilms and persister cells, highlighting its potential to overcome recalcitrant infections. BFU analysis further revealed that a statistically significant reduction in biofilm biomass was only observed at the MIC level (4 µg/mL), suggesting that the apparent inhibition of biofilm formation is largely attributable to suppression of planktonic growth rather than a direct interference with initial adhesion or matrix production. This observation underscores that the more clinically relevant property of BUN lies in its ability to eradicate mature biofilms, as supported by our biofilm eradication and persister cell killing assays. Given the observed biofilm eradication efficacy, BUN may effectively penetrate the extracellular polymeric substances (EPS) matrix within biofilms, possibly due to its amphiphilic structure and lipophilic tail that facilitate diffusion through the viscous polysaccharide barrier. Unlike enzymes or agents requiring prior degradation of the EPS matrix, BUN likely bypasses this barrier via direct membrane targeting and physicochemical penetration. This ability to access and disrupt deeply embedded bacterial cells may contribute to its superior biofilm eradication efficacy, as evidenced by our confocal imaging and CFU enumeration assays.

A low potential of BUN to induce bacterial resistance was observed. This contrasts with conventional antibiotics, such as vancomycin, which are prone to rapid resistance development due to well-documented mechanisms, including the acquisition of *vanA* and *vanB* gene clusters that alter peptidoglycan precursors, reducing vancomycin binding affinity (53). Additionally, prolonged vancomycin exposure can lead to stepwise resistance accumulation, further complicating treatment (54). In contrast, BUN maintained stable antimicrobial activity over multiple passages, with no significant increase

**TABLE 1** Antimicrobial susceptibility of BUN against enterococci[a]

| Strains | Characteristics | MIC (µg/mL) | MBC (µg/mL) |
|---|---|---|---|
| *E. faecalis* | | | |
| ATCC 29212 | Type strain, VSE | 4 | 4 |
| ATCC 51299 | Type strain, VRE | 4 | 4 |
| EFA2433084 | Clinical isolate, VSE | 4 | 4 |
| EFA2433159 | Clinical isolate, VSE | 4 | 4 |
| EFA2433254 | Clinical isolate, VSE | 4 | 8 |
| EFA24330726 | Clinical isolate, VSE | 4 | 8 |
| *E. faecium* | | | |
| ATCC 19434 | Type strain, VSE | 4 | 4 |
| U101 | Clinical isolate, VRE | 2 | 2 |
| SRYEFM1 | Clinical isolate, VRE | 2 | 4 |
| SRYEFM13 | Clinical isolate, VRE | 2 | 8 |
| SRYEFM2 | Clinical isolate, VRE | 2 | 2 |
| SRYEFM5 | Clinical isolate, VRE | 2 | 4 |
| SRYEFM9 | Clinical isolate, VRE | 2 | 2 |
| SRYEFM6 | Clinical isolate, VRE | 2 | 2 |
| SRYEFM15 | Clinical isolate, VRE | 2 | 2 |

[a]VSE: vancomycin-susceptible enterococci, VRE: vancomycin-resistant enterococci.

in MIC values, suggesting a lower propensity for resistance selection. This characteristic makes BUN a promising candidate for long-term treatment regimens. Moreover, the prolonged PAE observed in BUN further supports its potential for sustained antibacterial action, which could be particularly beneficial in clinical settings by allowing extended dosing intervals and reducing selective pressure for resistance.

Drug repurposing has become an increasingly important strategy in the search for new antibacterial agents, particularly against multidrug-resistant pathogens (55, 56). Within this framework, antiparasitic agents stand out as a promising subclass because of their structural diversity, intrinsic bioactivity, and frequent targeting of bacterial membranes or metabolic pathways—mechanistic features that overlap with those exploited by successful antibacterial drugs. Several representatives illustrate this point: tafenoquine displays membrane-targeting activity against MRSA; niclosamide disrupts bacterial energy metabolism; oxyclozanide and closantel, two salicylanilide-based anthelmintics, cause membrane depolarization in Gram-positive bacteria; and nitazoxanide inhibits anaerobic metabolism and alters membrane function.

BUN fits squarely within this mechanistic space but stands out for its rapid and robust bactericidal activity driven by direct membrane disruption. Unlike many other antiparasitic agents whose antibacterial effects are partly indirect or dependent on specific metabolic conditions, BUN acts swiftly and consistently across diverse settings. This unique combination of rapid killing and well-defined mechanism, together with its demonstrated *in vivo* efficacy against VRE, positions BUN as both a strong individual repurposing candidate and a representative model for exploring membrane-targeting antiparasitic agents as a broader pharmacological subclass.

It is also important to distinguish BUN from other nonantibiotic repurposed agents whose antimicrobial effects rely primarily on indirect mechanisms. For example, gallium compounds interfere with bacterial iron metabolism, depriving pathogens of essential nutrients (57), while disulfiram disrupts bacterial thiol metabolism, causing redox imbalance and oxidative stress (58). Such indirect approaches, while effective, often require additional host metabolic activation or specific environmental conditions to achieve optimal antibacterial effects. By contrast, BUN's direct membrane-disruptive action provides a reliable and immediate bactericidal effect across diverse conditions.

These findings suggest that BUN primarily targets the bacterial membrane, causing irreversible damage that leads to bacterial death. Experimental evidence, including lipid antagonism assays, demonstrated that BUN interacts specifically with PG, a key phospholipid component of the VRE membrane. In addition, MD simulations provided mechanistic insights into the interaction between BUN and bacterial membranes, revealing that BUN preferentially localizes to bacterial membrane phospholipids, leading to significant membrane destabilization and disruption. Membrane-targeting antibiotics offer several advantages over conventional intracellular-targeting antibiotics. First, since bacterial membranes are structurally distinct from mammalian membranes, these antibiotics can achieve selective bacterial killing while minimizing host cell toxicity (18). Second, membrane disruption leads to rapid bactericidal activity (59, 60). Third, bacteria have a lower propensity to develop resistance against membrane-targeting antibiotics compared to conventional antibiotics (61, 62). Compared to antimicrobial peptides (AMPs), which often suffer from poor stability, high production cost, and susceptibility to proteolytic degradation *in vivo*, BUN offers several practical advantages. As a small molecule with a defined chemical structure, BUN is more stable, easier to synthesize, and less likely to induce immune responses. Furthermore, while AMPs can sometimes act non-specifically and damage host cells, BUN showed selective membrane-targeting activity with minimal cytotoxicity in both *in vitro* and *in vivo* assays. Our findings highlight BUN's potential as a membrane-targeting antibacterial agent with potent bactericidal activity, low resistance induction, and sustained efficacy.

The subcutaneous abscess model was selected to assess the *in vivo* antimicrobial activity by BUN because soft tissue abscesses and deep-seated skin infections caused by VRE are clinically challenging due to bacterial persistence and poor antibiotic

penetration into infected tissue (8, 51). An abscess is an enclosed infection site, where biofilm formation and persister cells may contribute to antimicrobial resistance and treatment failure (63, 64). These infections are particularly difficult to manage because the biofilm matrix serves as a protective barrier, limiting antibiotic diffusion and immune system clearance. (65) In addition, we utilized the wound infection model to further evaluate the *in vivo* antimicrobial efficacy of BUN. Chronic wound infections, including surgical site infections, diabetic foot ulcers, and pressure sores, are common complications in immunocompromised patients and those with prolonged hospitalization (66). VRE is frequently associated with these infections, where its ability to persist, form biofilms, and delay wound healing exacerbates treatment difficulties (67). The observed reduction in bacterial load and accelerated wound healing in the BUN-treated mice suggests that BUN may be beneficial for managing VRE-associated wound infections, particularly in cases where conventional antibiotics often fail (68, 69). Together, these two *in vivo* models provide clinically relevant evidence supporting the potential of BUN as an antimicrobial agent for treating both soft tissue and wound infections caused by VRE. These findings highlight BUN's ability to target bacterial persistence and biofilms, key factors in recalcitrant VRE infections. However, further studies are still warranted to optimize BUN's PK and determine its efficacy in additional *in vivo* infection models (70).

A comprehensive toxicological assessment is crucial for evaluating the clinical potential of BUN. *In vitro* cytotoxicity studies demonstrated the good biocompatibility of BUN at therapeutic concentrations, which could probably be due to its specific interaction with bacterial cell membrane phospholipids. In consistence, the *in vivo* toxicity assay also showed that BUN did not cause organ damage with normal histopathological performance and blood-related biomarkers. These findings indicate that BUN exhibited a favorable safety profile under the tested conditions, although further investigations are required to characterize its PK properties and long-term safety across different administration routes. Remarkably, our *in vitro* apoptosis analysis employed a 24-h exposure duration, which reflects a standard acute toxicity assessment commonly adopted for preliminary screening. While this timeframe provides meaningful insights into short-term cytocompatibility, it may not fully capture the potential effects of prolonged or repeated drug exposure. Therefore, future studies involving extended incubation periods or chronic dosing models would be valuable to comprehensively evaluate BUN's long-term cytotoxicity and safety in human cells (71).

Despite its potent antimicrobial activity and favorable *in vivo* efficacy in the wound infection model, the PK profile of BUN suggests potential limitations for systemic therapy. In our study, BUN exhibited relatively rapid plasma clearance and only moderate systemic exposure, which could restrict its therapeutic window for sustained bactericidal activity. Such characteristics may necessitate optimized dosing regimens or advanced formulation strategies (e.g., sustained-release systems or targeted delivery) to achieve adequate systemic concentrations. Furthermore, the observed mechanistic differences compared with classical pore-forming agents may account for certain unexpected probe-specific responses, indicating that BUN disrupts bacterial membranes through progressive depolarization and permeability alteration rather than immediate large-pore formation. These aspects warrant further investigation to facilitate their potential clinical application.

In conclusion, BUN represents a promising therapeutic candidate for VRE infections, with potent bactericidal and antibiofilm activities. Its low potential for resistance induction, favorable cytotoxicity profile, and effective *in vivo* antibacterial activity position it as a strong contender for development into a clinical antimicrobial agent. Further studies are needed to optimize its PK and evaluate its safety and efficacy in clinical practice. Overall, BUN holds great promise as a therapeutic agent against VRE-related infections.

## ACKNOWLEDGMENTS

This study was supported by the National Natural Science Foundation of China (grant number: 82202591), the Natural Science Foundation of Hunan Province (grant numbers: 2023JJ40066, 2024JJ5514, and 2024JJ6081), and the "co‑PI" project from the Third Xiangya Hospital of Central South University (grant number: 202420).

P.S. and G.H. contributed equally to this work and shared the first authorship. P.S. and Y.W. designed the research; P.S. and G.H. conducted the experiments; P.S., G.H., and Y.W. analyzed the research results; S.G., D.X., Y.L., M.L., L.L., Y.H., Y.L., and L.Z. conducted the data collection; P.S., G.H., and Y.W. wrote and edited the manuscript. All authors have read and approved the submitted version of the manuscript.

## AUTHOR AFFILIATIONS

[1]Department of Laboratory Medicine, The Third Xiangya Hospital of Central South University, Changsha, Hunan, China

[2]Department of Laboratory Medicine, The Affiliated Changsha Hospital of Xiangya School of Medicine (The First Hospital of Changsha), Central South University, Changsha, Hunan, China

## AUTHOR ORCIDs

Yong Wu  http://orcid.org/0000-0002-3667-8716

## FUNDING

| Funder | Grant(s) | Author(s) |
|---|---|---|
| National Natural Science Foundation of China | 82202591 | Yong Wu |
| Natural Science Foundation of Hunan Province | 2024JJ5514 | Pengfei She |
| Natural Science Foundation of Hunan Province | 2024JJ6081 | Yimin Li |
| Natural Science Foundation of Hunan Province | 2023JJ40066 | Linying Zhou |

## AUTHOR CONTRIBUTIONS

Pengfei She, Conceptualization, Data curation, Formal analysis, Investigation, Methodology, Validation, Visualization, Writing – original draft, Writing – review and editing | Guanqing Huang, Conceptualization, Data curation, Formal analysis, Investigation, Methodology, Validation, Visualization, Writing – original draft, Writing – review and editing | Shaowei Guo, Data curation, Resources | Yiqing Liu, Data curation, Resources | Mengna Li, Data curation, Resources | Lihua Lu, Data curation, Resources | Yelan Hong, Data curation, Resources | Yimin Li, Funding acquisition, Resources | Linying Zhou, Funding acquisition, Resources.

## DATA AVAILABILITY

All data supporting the findings of this study are openly available in the Zenodo repository under the Creative Commons Attribution 4.0 International (CC BY 4.0) license. The data set includes raw values underlying means and standard deviations, statistical analysis results, and source figure files (Fig. 1 to 6 and Fig. S1 to S5). The data set is accessible at https://doi.org/10.5281/zenodo.15805153. The data that support the findings of this study are available from the corresponding author upon reasonable request.

## ETHICS APPROVAL

The animal study was approved by the Third Xiangya Hospital of Central South University (no. CSU-2024-0280).

## ADDITIONAL FILES

The following material is available online.

### Supplemental Material

**Supplemental material (mSystems00425-25-s0001.docx).** Supplemental figures and tables.

### Open Peer Review

**PEER REVIEW HISTORY (review-history.pdf).** An accounting of the reviewer comments and feedback.

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
