## [Reviewer comments · mSystems]

Repurposing Bunamidine Hydrochloride as a Potent Antimicrobial Agent Targeting Vancomycin-Resistant Enterococci Membranes

Yong Wu, Pengfei She, Guanqing Huang, Shaowei Guo, Dan Xiao, Yiqing Liu, Mengna Li, Lihua Lu, Yelan Hong, Yimin Li, and Linying Zhou

Corresponding Author(s): Yong Wu, The Affiliated Changsha Hospital of Xiangya School of Medicine, Central South University

Review Timeline:

Submission Date:	March 25, 2025
Editorial Decision:	July 18, 2025
Revision Received:	August 7, 2025
Editorial Decision:	September 19, 2025
Revision Received:	September 19, 2025
Accepted:	October 5, 2025

Editor: Li Cui

Reviewer(s): Disclosure of reviewer identity is with reference to reviewer comments included in decision letter(s). The following individuals involved in review of your submission have agreed to reveal their identity: Feifei Chen (Reviewer #1); Wenfang Lin (Reviewer #2)

Transaction Report:

DOI: <https://doi.org/10.1128/msystems.00425-25>

Re: mSystems00425-25 (**Repurposing Bunamidine Hydrochloride as a Potent Antimicrobial Agent Targeting Vancomycin-Resistant Enterococci Membranes**)

Dear Prof. Yong Wu:

Thank you for the privilege of reviewing your work. Below you will find my comments, instructions from the mSystems editorial office, and the reviewer comments. We apologize for the delay in the review process. Because only one reviewer was secured, I have reviewed this work and find it to be a systematic and valuable contribution. However, I agree with the reviewer's comments that further revisions are needed to improve the clarity and impact of this work.

Revision Guidelines

Sincerely,
Li Cui
Editor
mSystems

My comments:

Introduction: more information about the present progress on drug repurposing research should be added, especially on the class of antiparasitic agent.

Line 391, for the effect of BUN on biofilm-embedded persister cells, it is not clear how the persister cells were defined here, how

they were measured and how they were distinguished from growing cells.

Discussion: are there other antiparasitic agents in addition to BUN, is it possible to expand the findings to other antiparasitic agents or similar agents.

Reviewer #1 (Comments for the Author):

This study demonstrates the promising potential of bunamidine hydrochloride (BUN), a repurposed antiparasitic drug, as a potent membrane-targeting antimicrobial agent against vancomycin-resistant Enterococci (VRE). Through comprehensive in vitro and in vivo experiments, BUN exhibited rapid bactericidal activity, effective biofilm inhibition, and eradication of persister cells by selectively disrupting bacterial membrane integrity via interaction with phosphatidylglycerol (PG). While further mechanistic and pharmacokinetic studies are warranted, these findings highlight BUN as a promising therapeutic candidate for multidrug-resistant VRE infections, leveraging drug repurposing to address urgent clinical challenges.

Major concerns:

1 The study focuses on BUN's efficacy against Enterococci, but its activity against other clinically relevant Gram-positive and Gram-negative pathogens remains unexplored. Expanding this evaluation would better establish BUN's broad-spectrum potential.

2 Before assessing biofilm biomass (Fig. 2A & 2B), bacterial growth inhibition (OD600) should be measured, and biofilm formation units (BFUs = A560/A600) should be calculated for normalization. Additionally, the current images are unclear and should be improved for better resolution.

3 The PAE methodology is not described, and the MICs of VAN and DAP against the tested strains (*E. faecalis* ATCC 51299 and *E. faecium* U101) are missing. These details are essential for comparative analysis.

4 The positive control (MLT) showed strong PI fluorescence (Fig. 4E) but weaker SYTOX Green and DiSC3(5) signals. The authors should discuss possible mechanistic differences (e.g., pore formation vs. depolarization) that could explain this variation.

5 The IC50 values for HSF and HaCaT cells (Fig. 5A) may be unreliable, as the lowest viability levels (~60% and ~30%) did not reach sufficient inhibition for precise extrapolation. Testing higher concentrations would improve accuracy.

6 Fig. 4N should explicitly depict hydrogen bonds and hydrophobic interactions (e.g., dashed lines) between BUN and the 7DOPC/3DOPG membrane to provide clearer structural insights beyond Fig. 4M.

7 The discussion should address BUN's potential drawbacks, such as rapid clearance and limited systemic exposure, which could impact clinical translation. Additionally, any unexpected findings warrant mechanistic discussion to strengthen the study's robustness.

Minor concerns:

1 Lines 62 and 99: "Enterococci" should be italicized. Similar modifications should be made elsewhere if necessary.

2 Electron Microscopy vs. Cell Viability: 5×MIC BUN was used for electron microscopy, but only 1×MIC BUN was used for cell viability determination. Why chose different concentrations for these two experiments?

3 Line 250: The concentration of LP is missing.

4 Line 329: It appears to refer to Fig.1E instead of 1D.

5 In mouse wound infection model: Each group included n=6 mice. On days 1, 3, and 7, mice were sampled for bacterial burden counts, and additional samples were used for histological examination. The question is: Given only 6 mice per group, how were the experiments conducted? How many mice were sampled per time point?

6 The figures are currently arranged randomly. Please adjust their order to match their appearance in the text.

7 The statement "Different from bacteriostatic agents (such as Lusutrombopag)..." requires a reference to support this description.

8 The scale bar of Fig. 1F is missing in the microscopy images. Also, The scale bars are extremely blurry and illegible for Fig. 2C&2E.

9 The bacterial strain used in the CLSM experiment is not specified in the figure legend.

10 What does MLT represent in Figs. 4B, 4D, and 4E?

11 The pharmacokinetic (PK) study methods are not described, including which species (e.g., mice, rats) was used.

12 In Line 577, the phrase "a more significant reduction..." suggests a comparative effect. However, statistical analysis was not provided for Fig. 6D to confirm significance.

Dear Editor and Reviewer,

We sincerely thank you for the thorough and constructive comments, which have helped us substantially improve the clarity, rigor, and completeness of our manuscript. We have carefully addressed each point raised, as detailed below, and revised the manuscript accordingly. All changes are marked in the revised manuscript, with page and line numbers indicated in each response.

Editor's comments:

Comment 1

Introduction: Please supplement the current progress on drug repurposing research, particularly emphasizing studies involving the class of antiparasitic agents.

Response:

We thank the reviewer for this valuable suggestion. In the revised manuscript, we have expanded the Introduction section to provide a broader context on drug repurposing research, with a particular emphasis on antiparasitic agents. Specifically, we summarized some representative examples, including niclosamide, tafenoquine, oxyclozanide, closantel, and nitazoxanide, highlighting their reported antibacterial activities and underlying mechanisms (such as membrane targeting and metabolic disruption, etc). These additions place our study within the framework of recent advances in antimicrobial repurposing and underscore the rationale for investigating bunamidine hydrochloride (BUN) as a membrane-active antibacterial candidate.

Location in revised manuscript: Introduction, (Page 3, Lines 91–108)

Comment 2

Line 391: For the effect of BUN on biofilm-embedded persister cells, it is not clear how the persister cells were defined here, how they were measured, and how they were distinguished from growing cells.

Response:

We thank the reviewer for raising this important point. In our study, biofilm-embedded persister cells were operationally defined as metabolically quiescent, non-dividing bacterial cells that survive lethal antibiotic exposure within a mature biofilm matrix, consistent with widely accepted definitions in the literature (e.g., Conlon et al., 2016; de Breij et al., 2018). These cells are distinct from actively growing planktonic bacteria and are characterized by a transient, phenotypic tolerance to antibiotics rather than stable genetic resistance.

To enrich for persister cells, we adopted a rifampin-based induction protocol that has been validated in high-impact studies, including the Science Translational Medicine report by de Breij et al. (Sci. Transl. Med. 2018; 10: eaan4044), which described the

antimicrobial peptide SAAP-148. In brief, 24 h pre-formed *E. faecalis* biofilms were treated with a high dose of rifampin (100× MIC) under nutrient-limited conditions for 24 h. This treatment selectively eradicates metabolically active cells while allowing dormant persisters to survive. Planktonic cells were removed by PBS washing, and residual adherent persister cells within the biofilm were recovered by sonication. These cells were then used in time-kill assays under non-growth conditions (PBS, no nutrient supplementation) with BUN or comparator antibiotics (VAN, DAP).

This approach ensures that:

Source of cells — Only bacteria embedded within a pre-formed biofilm are analyzed, excluding the proliferating planktonic cells.

Distinction from growing cells — The absence of fresh nutrients during induction and killing assays minimizes regrowth and maintains cells in a dormant state.

Measurement — CFU enumeration at defined time points reflects killing efficiency specifically against the persister population, not against actively dividing cells.

By explicitly incorporating these steps into the Methods section (“Time-kill Kinetics of Biofilm-embedded Persister Cells”) (Page 7, Lines 259–272), we have clarified how persister cells were generated, measured, and distinguished from growing bacteria, in direct alignment with established protocols in the persister research field.

References:

1. Conlon, B. P., Rowe, S. E., Gandt, A. B., Nuxoll, A. S., Donegan, N. P., Zalis, E. A., ... & Lewis, K. (2016). Persister formation in *Staphylococcus aureus* is associated with ATP depletion. *Nature Microbiology*, 1, 16051. doi:10.1038/nmicrobiol.2016.51

2. de Breij, A., Riool, M., Cordfunke, R. A., Malanovic, N., de Boer, L., Koning, R. I., ... & Zaat, S. A. (2018). The antimicrobial peptide SAAP-148 combats drug-resistant bacteria and biofilms. *Science Translational Medicine*, 10(423), eaan4044. doi:[10.1126/scitranslmed.aan4044](https://doi.org/10.1126/scitranslmed.aan4044)

Comment 3

Discussion: Are there other antiparasitic agents in addition to BUN, and is it possible to expand the findings to other antiparasitic agents or similar compounds?

Response:

We appreciate the reviewer’s insightful question, which prompted us to place our

findings within the broader framework of antimicrobial drug repurposing. Drug repurposing has emerged as a strategic approach to address the urgent challenge of antimicrobial resistance, enabling the rapid identification of novel antibacterial activities from approved drugs with known pharmacological and safety profiles. Within this context, antiparasitic agents represent a particularly promising subclass owing to their structural diversity, intrinsic bioactivity, and frequent targeting of bacterial membranes or metabolic pathways—mechanistic features that overlap with those of successful antibacterial agents.

Representative examples underscore this potential: tafenoquine exhibits potent membrane-targeting activity against MRSA; niclosamide disrupts bacterial energy metabolism; oxyclozanide and closantel cause membrane depolarization in Gram-positive bacteria; and nitazoxanide inhibits anaerobic metabolism and alters membrane function. Despite differences in structure and specific targets, these antiparasitic compounds converge on the ability to compromise bacterial membrane integrity or disrupt essential energy systems, suggesting that membrane-active antiparasitic agents may constitute a repurposing pharmacological subclass with consistent antibacterial potential.

BUN aligns with this subclass yet stands out for its direct membrane-disruptive mechanism, which produces rapid and reliable bactericidal effects. Unlike some antiparasitic agents whose antibacterial activity is partly indirect or dependent on specific metabolic environments, BUN acts swiftly and consistently, as confirmed in both experimental and computational analyses, and demonstrates clear *in vivo* efficacy against VRE. This combination of rapid killing, defined mechanism, and translational potential supports BUN not only as an individual candidate for repurposing but also as a representative model for exploring membrane-targeting antiparasitic agents more broadly. We have incorporated this expanded discussion into the revised manuscript to better contextualize BUN within the wider landscape of antimicrobial drug repurposing.

Location in revised manuscript: Discussion section (Page 17, Lines 640–656).

Reviewer #1 (Comments for the Author):

This study demonstrates the promising potential of bunamidine hydrochloride (BUN), a repurposed antiparasitic drug, as a potent membrane-targeting antimicrobial agent against vancomycin-resistant Enterococci (VRE). Through comprehensive *in vitro* and *in vivo* experiments, BUN exhibited rapid bactericidal activity, effective biofilm inhibition, and eradication of persister cells by selectively disrupting bacterial membrane integrity via interaction with phosphatidylglycerol (PG). While further mechanistic and pharmacokinetic studies are warranted, these findings highlight BUN as a promising therapeutic candidate for

multidrug-resistant VRE infections, leveraging drug repurposing to address urgent clinical challenges.

Major Concerns

1. The study focuses on BUN's efficacy against Enterococci, but its activity against other clinically relevant Gram-positive and Gram-negative pathogens remains unexplored. Expanding this evaluation would better establish BUN's broad-spectrum potential.

Response:

We thank the reviewer for this valuable suggestion. In fact, BUN's antibacterial activity had previously been evaluated in our laboratory against a broad panel of clinically relevant Gram-positive and Gram-negative pathogens. However, due to space limitations, these results were not included in the original submission. Following the reviewer's advice, we now present representative MIC data for each major pathogen group, including methicillin-resistant *Staphylococcus aureus* (MRSA), *S. epidermidis* (SE⁺), *Escherichia coli*, *Acinetobacter baumannii*, *Pseudomonas aeruginosa*, and *Klebsiella pneumoniae*, along with the *Enterococcus* strains. These data are summarized in **Supplementary Table 2** and have been referenced in the revised Results section (Page 11, Lines 391–399).

2. Before assessing biofilm biomass (Fig. 2A & 2B), bacterial growth inhibition (OD₆₀₀) should be measured, and biofilm formation units (BFUs = A₅₆₀/A₆₀₀) should be calculated for normalization. Additionally, the current images are unclear and should be improved for better resolution.

Response:

We appreciate the reviewer's insightful suggestion to normalize biofilm biomass to account for differences in planktonic growth. In the revised study, we repeated the biofilm inhibition assay and measured planktonic growth (A_{600nm}) in parallel. Biofilm formation units (BFUs, A₅₆₀/A₆₀₀) were calculated, and the results are presented in Supplementary Figure S2. Our data indicate that BUN primarily inhibits biofilm formation by suppressing planktonic bacterial growth, as both biomass reduction and BFU changes became significant only at $\geq 4 \mu\text{g/mL}$.

We fully acknowledge the reviewer's point and have incorporated normalized BFU data to provide a more accurate interpretation. Nevertheless, given that biofilm eradication has greater clinical relevance than biofilm inhibition, and that BUN demonstrated potent biofilm eradicating activity against pre-formed biofilms (Figures 2B, 2F, 2I), our study emphasizes the biofilm eradication activity of BUN as a key focus.

Location in revised manuscript: Discussion, (Page 16, Lines 618–624)

3. The PAE methodology is not described, and the MICs of VAN and DAP against the tested strains (*E. faecalis* ATCC 51299 and *E. faecium* U101) are missing. These details are essential for comparative analysis.

Response:

We have now added a detailed description of the PAE assay in the Methods section (Page 5, Lines 169–178). The MICs of vancomycin (VAN) and daptomycin (DAP) against both *E. faecalis* ATCC 51299 and *E. faecium* U101 were determined to be 32 µg/mL and 4 µg/mL, respectively (Page 5, Lines 178–181). Unless otherwise specified, VAN and DAP were used at 10× MIC in all relevant comparative assays, including Figures 2K and 3D.

4. The positive control (MLT) showed strong PI fluorescence (Fig. 4E) but weaker SYTOX Green and DiSC3(5) signals. The authors should discuss possible mechanistic differences (e.g., pore formation vs. depolarization) that could explain this variation.

Response:

We appreciate this insightful comment. We have now included an expanded discussion in the revised manuscript (Page 13–14, Lines 499–506) addressing potential mechanistic differences. Specifically, MLT may primarily induce large membrane pores that facilitate PI penetration but have a lesser effect on membrane potential, leading to weaker DiSC3(5) responses. This distinction has been clarified in the Discussion.

5. The IC₅₀ values for HSF and HaCaT cells (Fig. 5A) may be unreliable, as the lowest viability levels (~60% and ~30%) did not reach sufficient inhibition for precise extrapolation. Testing higher concentrations would improve accuracy.

Response:

We appreciate the reviewer's concern regarding the accuracy of the IC₅₀ estimation. In the original experiment, the tested concentration range (0–32 µg/mL) did not achieve complete growth inhibition, which could affect curve fitting. Following the reviewer's suggestion, we have extended the concentration range up to 256 µg/mL for both HSF and HaCaT cells. The revised CCK-8 assay results now cover a full dose–response range, allowing more reliable IC₅₀ calculation. Updated data and curves have been incorporated into Figure 5A, and the Methods section has been modified accordingly (Page 6, Lines 214–226).

6. Fig. 4N should explicitly depict hydrogen bonds and hydrophobic interactions (e.g., dashed lines) between BUN and the 7DOPC/3DOPG membrane to provide clearer structural insights beyond Fig. 4M.

Response:

We thank the reviewer for this valuable suggestion. Our molecular dynamics (MD)

simulations demonstrated that BUN binds to the 7DOPC/3DOPG bilayer primarily through diffuse hydrophobic contacts between its aromatic/alkyl moieties and the lipid acyl chains. Hydrogen bonding events were observed to be rare, transient, and contributed minimally to the overall binding stability.

Unlike hydrogen bonds, which possess discrete geometric constraints and can be represented unambiguously, hydrophobic interactions are non-directional, spatially diffuse, and often involve multiple lipid chains simultaneously. This makes their precise graphical depiction inherently ambiguous and at risk of overinterpretation. For this reason, and in line with common structural visualization practices, we have opted not to annotate individual hydrophobic contacts or hydrogen bonds in Fig. 4N.

Instead, the figure is intended to illustrate the overall spatial orientation of BUN relative to the membrane surface, without implying specific, static interaction sites. We have clarified this in both the Methods section (Page 8, Lines 299–305) and the Figure 4N legend (Page 26, Lines 1024–1026), explicitly noting that individual hydrophobic and hydrogen bonding contacts are not marked to maintain scientific accuracy and avoid misrepresentation.

7. The discussion should address BUN's potential drawbacks, such as rapid clearance and limited systemic exposure, which could impact clinical translation. Additionally, any unexpected findings warrant mechanistic discussion to strengthen the study's robustness.

Response:

We agree that it is important to discuss potential pharmacokinetic limitations. In the revised Discussion (Page 18–19, Lines 713–724), we have included a new paragraph noting that BUN may have relatively rapid clearance and moderate bioavailability, which could limit systemic exposure. These characteristics may necessitate optimized dosing regimens, formulation strategies, or combination therapy for effective systemic use. We also address unexpected observations, such as differential activity against persister cells compared to conventional agents, and propose possible mechanistic explanations. This addition provides a more balanced perspective on BUN's therapeutic potential and limitations.

Minor Concerns

1. Lines 62 and 99: "Enterococci" should be italicized.

Response: We thank the reviewer for pointing this out. We have carefully checked the manuscript and italicized *Enterococci* throughout, including the instances in Lines 62 and 117, as well as other occurrences where necessary.

2. Electron Microscopy vs. Cell Viability: 5×MIC BUN was used for electron microscopy, but only 1×MIC BUN was used for cell viability determination. Why

choose different concentrations for these two experiments?

Response: We appreciate this question. The electron microscopy assay was designed to visualize clear morphological alterations, which are more pronounced at higher concentrations (5×MIC). In contrast, the cell viability assay was intended to evaluate bactericidal efficacy under clinically relevant inhibitory concentrations (1×MIC).

3. Line 250: The concentration of LP is missing.

Response: We apologize for the omission. The concentration of Lusutrombopag (LP) used in the lysis assay was 32 µg/mL, the same as BUN. This information has now been added to the Methods section (Page 9, Lines 314–317).

4. Line 329: It appears to refer to Fig. 1E instead of 1D.

Response: Thank you for noticing this oversight. The figure reference has been corrected from Fig. 1D to Fig. 1E (Page 11, Lines 402–408).

5. Mouse wound infection model: Each group included n=6 mice. On days 1, 3, and 7, mice were sampled for bacterial burden counts, and additional samples were used for histological examination. Given only 6 mice per group, how were the experiments conducted? How many mice were sampled per time point?

Response: We apologize for the ambiguity in our original description. In fact, for each sampling time point (days 1, 3, and 7), an independent cohort of mice was used. Specifically, each treatment group contained nine mice per time point. Of these, six mice per group were sacrificed for bacterial burden quantification, and the remaining three mice per group were processed for histological examination at the corresponding time point. Thus, the mice used for bacterial counts and histology at different time points were from separate, dedicated cohorts, rather than being repeatedly sampled from the same animals. This clarification has been added to the Methods section (Page 9, Lines 327–336) to ensure accuracy and avoid misunderstanding.

6. The figures are currently arranged randomly. Please adjust their order to match their appearance in the text.

Response: We have reorganized the figures so that their numbering follows the sequence of appearance in the main text, ensuring consistency between figure order and manuscript flow.

Location in revised manuscript: Results (Page 13, Lines 494–499)

7. The statement "Different from bacteriostatic agents (such as Lusutrombopag)..." requires a reference to support this description.

Response: We have now cited the appropriate reference describing the antibacterial properties of Lusutrombopag to support this statement (Page 11, Lines 399–400).

8. The scale bar of Fig. 1F is missing in the microscopy images. Also, the scale bars are extremely blurry and illegible for Fig. 2C & 2E.

Response: We have added the missing scale bar to Fig. 1F and replaced the scale bars in Fig. 2C and Fig. 2E with clearer, high-resolution versions to improve legibility.

9. The bacterial strain used in the CLSM experiment is not specified in the figure legend.

Response: We have now specified the bacterial strain (*E. faecalis* ATCC 51299) in the legends for all relevant CLSM figures (Fig. 1F, Fig. 2)(Page 25, Lines 981–982; Lines 985–986) and Methods section (Page 5, Lines 154–155).

10. What does MLT represent in Figs. 4B, 4D, and 4E?

Response: MLT refers to melittin, a well-characterized pore-forming peptide used as a positive control in membrane-targeting assays. In this study, it was tested at 8 µg/mL, corresponding to its MIC against *E. faecalis* ATCC 51299, to provide a relevant bactericidal reference for comparison with BUN. This clarification has been added in the Methods section (Page 6, Lines 193–194) and its functional relevance is discussed in the Results section (Page 13–14, Lines 499–506).

11. The pharmacokinetic (PK) study methods are not described, including which species (e.g., mice, rats) was used.

Response: We have added a detailed description of the PK study methods in the Methods section, specifying that female ICR mice (6–8 weeks old, 23–27 g) were used, together with dosing regimen, sampling strategy, LC–MS/MS analytical conditions, and data analysis procedures (Page 10, Lines 364–377).

12. In Line 577, the phrase "a more significant reduction..." suggests a comparative effect. However, statistical analysis was not provided for Fig. 6D to confirm significance.

Response: We thank the reviewer for pointing out this important issue. To avoid overinterpretation, we have revised the related text in the Results to state that "BUN treatment showed a clearer downward trend in infection size compared with Vehicle" rather than implying statistical significance. We sincerely apologize for any confusion caused by the previous wording, and we have retained the original figure as it still effectively illustrates the comparative treatment trends over time.

Location in revised manuscript: Introduction section (Page15, Lines 583–585).

We sincerely appreciate your thorough and thoughtful review, which has undoubtedly enhanced the quality of this manuscript. We hope that the revisions we made have adequately addressed your concerns. If you have any further suggestions, please do not hesitate to let us know.

Once again, thank you for your time and expertise in reviewing our manuscript.

Best regards,

Wu Yong,

The Affiliated Changsha Hospital of Xiangya School of Medicine (The First Hospital of Changsha), Central South University, Changsha, Hunan, China.

Re: mSystems00425-25R1 (Repurposing Bunamidine Hydrochloride as a Potent Antimicrobial Agent Targeting Vancomycin-Resistant Enterococci Membranes)

Dear Prof. Yong Wu:

Revision Guidelines

Sincerely,
Li Cui
Editor
mSystems

Reviewer #2 (Comments for the Author):

General Comments:

This study investigated the inhibitory effect of bunamidine hydrochloride (BUN), a conventional antiparasitic agent, on clinically important vancomycin-resistant Enterococci. Interestingly, BUN exhibited strong bactericidal effects against both biofilm and persister cells. The underlying mechanisms related to membrane integrity were demonstrated through a series of experiments. Furthermore, the toxicity of BUN was validated using an in vivo murine infection model. Overall, the paper has a rigorous

experimental design, solid data, and in-depth mechanistic exploration.

However, it would be beneficial to test the antimicrobial effects of BUN on other representative Gram-positive bacteria, such as *Staphylococcus aureus*. Add more discussion on how BUN penetrates extracellular polymeric substances in biofilms to target the inner cells. Explain the advantage of BUN compared to antimicrobial peptides?

Details:

- 1.Line 66-72: Vancomycin is a last-resort antibiotic for resistant Gram-positive infections. Please elaborate on the clinical and public health significance of studying vancomycin-resistant bacteria.
- 2.Line 121: The authors selected only *Enterococcus* isolates. Would BUN be effective against other major Gram-positive bacteria such as *Staphylococcus aureus*? Expanding the bacterial range tested would be better.
- 3.Line 559: The concentration of BUN was $1 \times \text{MIC}$ with 24 h of incubation, which is relatively low considering the requirements for long-term treatment in real-world situations.
- 4.Figures: Figure 1G, Figure 2D and 2H, did have the Y-axis labeled.

Response to Reviewer #2

We sincerely thank you for your constructive comments and valuable suggestions. We appreciate your recognition of the rigorous experimental design and mechanistic exploration of our work. Below, we have addressed each point in detail and made corresponding revisions to improve the manuscript.

General Comment:

“It would be beneficial to test the antimicrobial effects of BUN on other representative Gram-positive bacteria, such as *Staphylococcus aureus*...”

Thank you for this insightful suggestion! Actually, we have tested the antimicrobial activity of BUN against several clinically relevant *Staphylococcus aureus* strains recent days, and we plan to further explore its systemic antimicrobial effects in vitro and in vivo and its underlying mechanisms. Of course, in the present manuscript, we can add some information about the anti-*S. aureus* effect by BUN. As shown in Supplementary Table 2, BUN exhibited comparable antimicrobial activity against varied *S. aureus* strains, including methicillin-resistant *S. aureus* (MRSA) strains of USA300, SAJ1, and SA1901, as well as a methicillin-susceptible *S. aureus* (MSSA) strain LZB1, with MIC values ranging from 4 to 8 $\mu\text{g/mL}$, which is similar to its activity against VRE isolates. These results suggest that BUN may possess broad antibacterial potential against Gram-positive bacteria beyond Enterococci. These data have been added to the Results section (Lines 392 - 398). However, considering the multidrug-resistant nature and high clinical burden of VRE, we retained our primary focus on *Enterococcus* spp. in this study.

“Add more discussion on how BUN penetrates extracellular polymeric substances in biofilms...”

Thank you for pointing this out. We have added further discussion in the revised Discussion section (Lines 624 - 632), elaborating on the potential mechanisms by which BUN may diffuse through the biofilm matrix, possibly via its amphiphilic structure and sustained membrane affinity, enabling deeper penetration and interaction with inner-layer persister cells. Although further work is warranted, our confocal microscopy and biomass eradication results (Figure 2) could partially support BUN's biofilm-penetrating capability.

“Explain the advantage of BUN compared to antimicrobial peptides?”

This is an excellent question. We have now addressed this point in the Discussion (Lines 684 - 690), emphasizing that BUN, as a small molecule with membrane-disruptive action, offers potential advantages over antimicrobial peptides (AMPs), including better chemical stability, cost-effective synthesis, and reduced

susceptibility to proteolytic degradation. These features may render BUN more suitable for clinical development and formulation.

Detailed Comments:

1. Line 66 – 72: “Please elaborate on the clinical and public health significance of studying vancomycin-resistant bacteria.”

Thank you for this suggestion. We have revised the Introduction (Lines 65 – 73) to emphasize the clinical and public health significance of vancomycin-resistant Enterococci (VRE), supported by several key studies:

“VRE infections are associated with prolonged hospitalization, increased medical costs, and up to 2.5-fold higher mortality [Mundy et al., 2000] . In immunocompromised patients, especially those with hematologic malignancies, VRE colonization markedly elevates the risk of bloodstream infection and death [Meschiari et al., 2023] . Moreover, *vanA/vanB* genes can transfer horizontally to *S. aureus*, facilitating the emergence of VRSA and further complicating clinical management [Girijan et al., 2021] .” These concerns highlight the urgent need to explore alternative anti-VRE strategies, including non-traditional agents like BUN.

2. Line 121: “Would BUN be effective against other major Gram-positive bacteria such as *Staphylococcus aureus*?”

As mentioned above, we have included MIC data of BUN against *S. aureus* and other Gram-positive pathogens in Supplementary Table 2 and related Results (Lines 392 – 398), addressing this point directly.

3. Line 559: “The concentration of BUN was $1 \times$ MIC with 24 h of incubation, which is relatively low considering the requirements for long-term treatment in real-world situations.”

Thank you for this valuable comment. We fully agree that prolonged or repeated exposure better reflects real-world treatment scenarios. In our study, the 24-hour incubation period used for the apoptosis assay followed internationally recognized acute cytotoxicity screening protocols, such as ISO 10993-5:2009, which is widely adopted for preliminary biocompatibility evaluations of medical materials.

Nevertheless, we acknowledge this represents only a short-term assessment. To more comprehensively evaluate BUN’s long-term cytotoxicity and safety under clinically relevant conditions, extended incubation periods or chronic exposure models will be incorporated in future studies. A clarification has been added in the Results section

(Lines 560 – 562), and the limitation is now explicitly discussed in the Discussion section (Lines 723 – 729).

4. Figures 1G, 2D, 2H: “Did have the Y-axis labeled.”

Thank you for pointing this out. We sincerely apologize for the oversight in the original submission, where we neglected to include the necessary color legends indicating SYTO (live) and PI (dead) staining in Figures 1G, 2D, and 2H. This omission may have caused confusion in interpreting the fluorescent signal proportions. We have now added clear vertical legends to each corresponding figure to explicitly label the stained populations. These improvements are shown in the revised figures and are also annotated in the Marked-up version of the manuscript (Lines 1031 – 1046).

Once again, we thank the reviewer for the insightful feedback that helped us improve the quality and clarity of our work. We hope the revised manuscript meets your expectations.

Best regards,

Wu Yong,

The Affiliated Changsha Hospital of Xiangya School of Medicine (The First Hospital of Changsha), Central South University, Changsha, Hunan, China.

Re: mSystems00425-25R2 (**Repurposing Bunamidine Hydrochloride as a Potent Antimicrobial Agent Targeting Vancomycin-Resistant Enterococci Membranes**)

Dear Prof. Yong Wu:

Your manuscript has been accepted, and I am forwarding it to the ASM production staff for publication. Your paper will first be checked to make sure all elements meet the technical requirements. ASM staff will contact you if anything needs to be revised before copyediting and production can begin. Otherwise, you will be notified when your proofs are ready to be viewed.

Sincerely,
Li Cui
Editor
mSystems

Reviewer #2 (Comments for the Author):

The author has replied to all my questions and provided reasonable explanations.